# Immediate neural impact and incomplete compensation after semantic hub disconnection

Zsuzsanna Kocsis [1,2,3,14] ✉, Rick L. Jenison [4], Peter N. Taylor[5,6], Ryan M. Calmus[1,2], Bob McMurray [7], Ariane E. Rhone[1], McCall E. Sarrett[8], Carolina Deifelt Streese[1], Yukiko Kikuchi [2], Phillip E. Gander[1,9,10], Joel I. Berger[1], Christopher K. Kovach [1], Inyong Choi[11], Jeremy D. Greenlee [1], Hiroto Kawasaki[1], Thomas E. Cope[12,13,14], Timothy D. Griffiths [2,14], Matthew A. Howard III[1,14] & Christopher I. Petkov[1,2,14] ✉

The human brain extracts meaning using an extensive neural system for semantic knowledge. Whether broadly distributed systems depend on or can compensate after losing a highly interconnected hub is controversial. We report intracranial recordings from two patients during a speech prediction task, obtained minutes before and after neurosurgical treatment requiring disconnection of the left anterior temporal lobe (ATL), a candidate semantic knowledge hub. Informed by modern diaschisis and predictive coding frameworks, we tested hypotheses ranging from solely neural network disruption to complete compensation by the indirectly affected language-related and speech-processing sites. Immediately after ATL disconnection, we observed neurophysiological alterations in the recorded frontal and auditory sites, providing direct evidence for the importance of the ATL as a semantic hub. We also obtained evidence for rapid, albeit incomplete, attempts at neural network compensation, with neural impact largely in the forms stipulated by the predictive coding framework, in specificity, and the modern diaschisis framework, more generally. The overall results validate these frameworks and reveal an immediate impact and capability of the human brain to adjust after losing a brain hub.

Physical and artificial networks are highly interconnected distributed systems. There is considerable interest in understanding how dense networks reorganize and compensate for system impact[1,2]. By some accounts, hubs−as highly interconnected network nodes−are instrumental for network function and their loss may not be recoverable[3,4]. By others, a distributed system may, at least in part, be able to compensate for losing a hub[5,6]. The strongest level of causal evidence for the role of a human brain hub is to evaluate its acute network-level impact following brain hub disconnection and any rapid functional compensation that ensues. This may depend on situations where

neurosurgical patient treatment can provide neurophysiological data shortly before and after the surgical procedure affecting a brain hub.

Speech is an important acoustic signal used by humans to communicate meaning. Successful comprehension of speech sounds engages a broad human brain network that rapidly decodes complex acoustic signals and binds them with categorical and contextual information crucial for concept formation and semantic knowledge[5–11]. By all accounts, this system is extensive. However, some accounts propose an indispensable role for the left anterior temporal lobe (ATL) as a hub for semantic knowledge, and others do not. A hub-and-spokes

model posits that the hub-like connectivity and function of the left ATL not only binds and establishes semantic representations in interaction with the other nodes or 'spokes' in the network, but it is important for efferent predictions about forthcoming speech signals influencing sensory cortex[4,12–16]. By this model, the specific loss of the left ATL would severely disrupt semantic processes, which is supported by patient studies, whereby stroke, surgical or degenerative damage to the ATL impairs responsive naming[17–19] and semantic memory[20–23] and reduces the efficiency of language processing[24,25]. By comparison, distributed only or multiple hub models posit that other regions could compensate for an impact on the ATL, or can maintain established semantic representations, at least temporarily[5,7,11,15]. Candidate compensatory regions would include the language-related hub in the left inferior frontal gyrus (IFG) and speech-processing temporal lobe regions[26]. However, as recently noted[15], a direct causal test of the functional relationship between the hub and the distributed nodes is lacking, and it remains unknown what the immediate site-specific neurophysiological impact on and potential compensation by a human neural network might be after the loss of one of its hubs.

In this study, we directly compared intracranial recordings obtained tens of minutes before and after a neurosurgical treatment that required disconnection of the left ATL (Fig. 1). These data are rare, collected in neurosurgery patients as part of an acute, one day left hemisphere ATL disconnection procedure to access and remove a temporal lobe seizure focus. Both patients had similar pre- and post-disconnection electrode coverage for clinical monitoring that included speech-responsive IFG and auditory cortical sites (Fig. 2), and they were tested awake as part of clinical language mapping during both recording periods. The data provide insights into how a neural network dynamically alters after an impact on the left ATL, with site-specific neurophysiological precision and immediacy rarely possible to obtain from the human brain. By comparison, most human neurosurgery patient studies can provide intracranial site-specific neurophysiological information prior to, but typically not after the surgical treatment. On the other hand, perioperative knowledge often stems primarily from noninvasive neuroimaging studies of functional and structural connectome effects days or weeks before and after the operation or a brain lesion[27]. Although animal models provide higher precision in studying hub-like connectivity and function at cellular levels[28], humans are the ideal model for language function. Guided by three sets of hypotheses, this study provides direct causal evidence

that implicates the ATL as a key hub for speech semantic processes, while also revealing a remarkably rapid capability of the distributed nodes to immediately adjust after an impact on a hub.

The possible acute impact on the fronto-temporal system following the loss of the left ATL can be characterized in terms of three overarching sets of hypotheses. 1) No impact hypothesis: The unaffected semantic network brain areas could at least temporarily stabilize speech responses and predictions, including the language relevant IFG (pars triangularis and opercularis) and speech processing areas in auditory cortex. Support for this hypothesis is expected in the form of minimal to no immediate change pre- versus post-disconnection in speech and speech prediction-related processes in the fronto-temporal network. This would include unaffected effective (directed) connectivity between the remaining nodes, provided that the ATL is not indispensable for these functions (illustrated in Fig. 1B left panel). Such a result would support the distributed only or multiple hub models whereby the role of the ATL can be immediately compensated for[5,7,15,29]. 2) Classical diaschisis or solely disruption hypothesis: Diaschisis, by its classical definition[30], is functional disruption by damage to an area affecting not only local but also distant interconnected sites[27,30,31]. A classical diaschisis hypothesis goes beyond the expected immediate impact on areas near the surgical site, predicting disruption at distant but interconnected fronto-temporal sites. Results supporting this hypothesis would include decreased responses to speech in IFG and auditory cortical regions and reduced effective connectivity after ATL disconnection (Fig. 1B center panel). Full support for a classical diaschisis hypothesis would demonstrate an irreplaceable role of the left ATL in this acute phase, supporting the hub-and-spokes model[4,15]. 3) Incomplete immediate compensation (predictive coding or modern diaschisis) hypothesis: Amidst disrupted function, there could be specific neurophysiological alterations that can be anticipated by, for instance, a 'predictive coding' framework (see next paragraph). More generally, this hypothesis aligns with a modern take on the diaschisis hypothesis[31], whereby functional abnormalities in distant interconnected sites can include greater excitability and increased functional connectivity and connectome effects. The neural network, for instance, may increase functional interconnectivity and its neural network properties change in an effort to compensate for the impact to the system[27,32,33]. Results supporting this hypothesis would go beyond simply disruption by showing evidence for specific alterations in

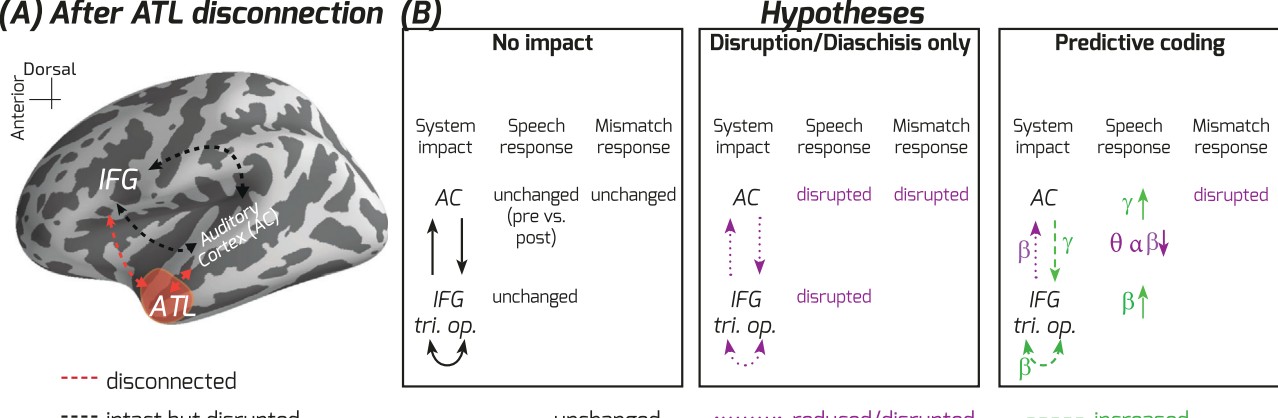

**Fig. 1 | Illustrated hypotheses. A** Left hemisphere ATL disconnection would result in a structural disconnection of pathways (red) interconnecting with the ATL but no direct impact on other pathways such as those interconnecting Auditory Cortex (AC) and Inferior Frontal Gyrus (IFG), regions well removed from the surgical site to access the seizure focus. **B** Left column: No impact hypothesis would result in no change after the ATL disconnection in AC and IFG effective connectivity, AC speech

and speech prediction mismatch responses. Middle column: Diaschisis or disruption-only hypothesis would result solely in disruption of ATL interconnected sites such as the AC and IFG. Right column: Predictive coding hypothesis posits specific patterns of disruption and compensation, including magnified high gamma responses to the target speech sound, a disrupted speech mismatch response in AC, and increased effective connectivity driven by the IFG.

**(A) ATL disconnection and resection**

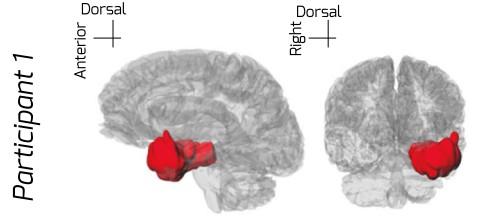
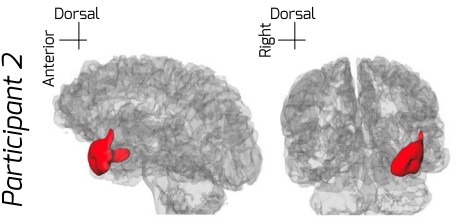

**(B) Electrode coverage**

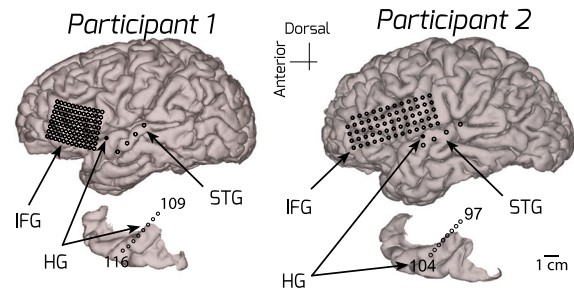
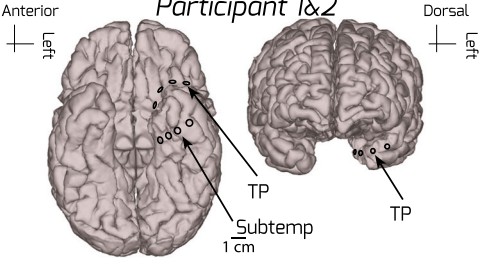

**(C) ATL disconnection connectome impact**

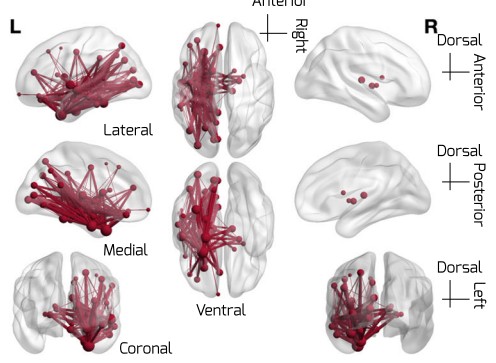

Participant 1

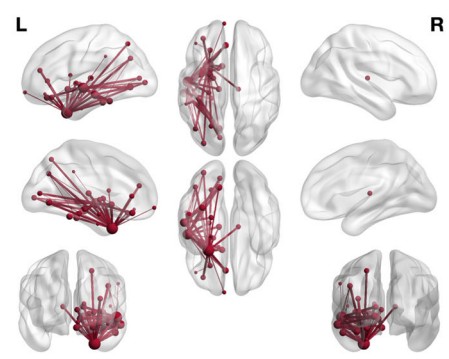

Participant 2

**(D) Speech prediction paradigm**

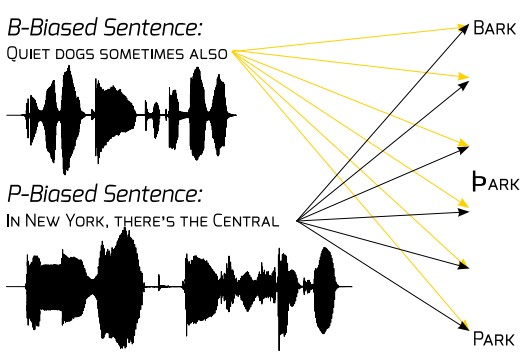

**(E) Behavioral results**

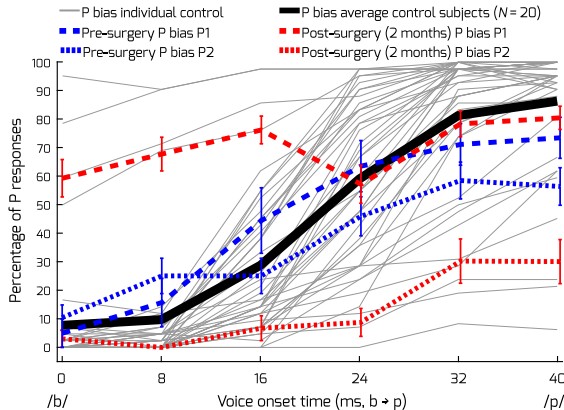

speech and speech prediction signals, focal increases in effective connectivity[12,13], or systematic changes in hub-like properties and function in the remaining network nodes after the loss of the ATL[34,35]. Support for this hypothesis would both underscore the important role of the left ATL[4,15] and highlight the capability for rapid systematic neurophysiological alterations by the distributed network nodes[5,7,29].

For the predictive coding hypothesis, we relied on the canonical microcircuit implementation of the predictive coding framework to generate more concrete predictions about both the form of disrupted and potentially compensatory site-specific neurophysiological responses and interconnectivity between the sites[36-38] (Fig. 1B right panel). Predictive coding theory posits that higher-order areas

**Fig. 2 | Participant electrode coverage, ATL disconnection structural impact, task, and behavior. A** Reconstructed surfaces showing the disconnected regions for each participant in red; P1's region also includes the MTL resection, which occurred after the ATL disconnection and recordings reported here (Supplementary Fig. 3). **B** Electrode coverage in the two participants. **C** Impact of ATL disconnection on structural connectivity pathways: Diffusion-weighted imaging derived networks show the reduction of streamlines caused by the ATL disconnection. Tube thickness is proportional to the percentage of streamlines removed from that connection. The size of each sphere represents the percentage reduction in the number of streamlines connecting that region, visualized using BrainNet Viewer[106]. **D** Depiction of the speech prediction paradigm. Participants listened to /b/ or /p/ word biasing sentences with the final target word manipulated along a /b/ to /p/ voice onset time (VOT) continuum. **E** Behavioral responses under the /p/ bias condition, as a function of the /b/ to /p/ VOT continuum and the percentage of P responses given by the participants. The thin gray lines show each control participant (individuals without epilepsy who performed the task, based on[48]) and the thick black line their average psychometric function. The blue dashed lines show the average responses of P1 and P2 before the surgery along with the standard error of mean (SEM) across trials, which fit within the normative distribution established by the control participants. The red dashed lines show the patients' average responses 2 months after their ATL disconnection surgery along with the standard error of mean across trials, showing substantial disruption of their /p/ bias psychometric functions. Only results for the /p/ biased sentences are shown here because, unlike the controls, both before and after disconnection the patients almost never responded having heard /p/ in the /b/ bias condition regardless of the VOT (Supplementary Fig. 1). HG Heschl's gyrus, ATL anterior temporal lobe, IFG inferior frontal gyrus, Subtemp subtemporal area, TP temporal pole.

generate top-down predictions that influence sensory cortical processing manifested in lower frequencies (e.g., beta band oscillations: 12–30 Hz)[36]. By this account, sensory cortex does not simply conduct sensory processes but can process differences between the top-down prediction and the sensory evidence. When predictions match the sensory evidence, there is very little prediction error fed forward from sensory cortex to higher-order areas. When the predictions do not match the sensory evidence, a prediction error is fed forward to update the network, which can be evident in two forms: mismatch responses to unexpected sensory stimuli and gamma activity in local field potentials (>30 Hz) associated with feed-forward neurophysiological signaling of the prediction error signal[36–38]. Moreover, higher-order areas are expected to feedback sensory inference (predictive) signals to sensory cortex in the form of lower frequency neurophysiological signals (e.g., beta/alpha, or <30 Hz)[39–42].

We summarize our specific predictive coding hypothesis predictions, as follows: First, when ATL speech predictions are affected by the loss of the ATL in a way that is not fully compensated for by the other areas, auditory cortex should generate abnormally magnified higher frequency responses to the speech sounds (e.g., gamma; >30 Hz)[36,40,43] and disrupted mismatch responses to unpredicted versus predicted speech sounds[44,45]. Second, there should be evidence of IFG and higher-order auditory cortical areas attempting to reinstate affected semantic predictions after the loss of the ATL in the form of increased low-frequency speech responses and interactions with auditory cortex, possibly involving lower frequency theta oscillations, rather than gamma, implicated in inter-regional interconnectivity and cognitive function[32,33,46,47]. As another vantage point on network impact (e.g., connectome diaschisis[31]) and possible compensation following ATL disconnection, we conducted graph theoretic analyses of hub centrality and network effective connectivity. In this case-of-cases study, we ensured that all main interpretations are based on significant and consistent effects in both participants. We obtained evidence for the modern diaschisis and predictive coding (incomplete immediate compensation) hypothesis, underscoring the importance of the ATL as a semantic hub in this acute phase.

## Results
### Post-surgical impact on behavioral semantic perception
To evaluate the post-surgical impact on speech predictions in the two patients, we employed an established task where higher-level sentence context influences speech perception and predictions[48,49]. The task was conducted with the two participants 2–6 weeks before and 2 months after the surgical procedure. It consisted of listening to sentences that biased the perception of a final target word. The task allowed us to assess both congruent semantic predictions (e.g., "In New York there's the Central park"; Fig. 2D) and incongruent conditions (e.g., "In New York there's the Central bark"). The voice onset time (VOT) of the target word was manipulated along a perceptual continuum from /b/ as in 'bark' (short VOTs) to /p/ as in 'park' (long

VOTs), with several different /b/ and /p/ word pairs being used. Thus, the preceding sentence context, alongside a simple VOT change in the first phoneme of otherwise acoustically matched target words, biases the semantic perception of the words.

Figure 2E shows that, before surgery, both participants' phoneme categorization functions displayed an initial semantic context effect that closely mirrored the average response to the /p/ biased condition in the control participants, with both blue dashed lines largely overlapping with the thick black line (controls). Two months after surgery, both participants had a much-reduced effect of voice onset time, manifest in flatter red dashed lines. Participant 1 adopted a strategy across both conditions (c.f. Supplementary Fig. 1) that was more heavily based upon preceding sentential context, with an overall shift upwards towards the /p/ bias. Conversely, Participant 2 retained some ability to integrate voice onset time, but this was diminished from the pre-surgical state, and they showed a new bias in the opposite direction towards /b/, incongruent with the preceding sentential context. This bias cannot be easily explained as a response button bias given their good performance on the control tasks (see below). Overall, while the post-surgical biases of the two patients on the semantic priming task appear quite different, they both represent a loss of adaptive function. This is analogous to the failure of comprehension resulting from overly precise perceptual predictions in patients with damage to frontal lobe language regions[49,50]. This was statistically confirmed by a logistic mixed model run separately for each patient which showed a significant or near significant main effect of Disconnection for both patients (Participant 1: $Z = 3.44$, $p < 0.001$; Participant 2: $Z = -1.95$, $p = 0.051$). The form of the post-surgical behavioral impact was different between the two subjects (See Supplementary Table 1 for more details and the complete results), potentially because of the more extensive additional medial temporal lobe resection in the case of Participant 1 (P1) conducted after the post-disconnection recordings.

Both participants performed very well on the control tasks in which there was no lexical competition, and the semantic context was never incongruent with the VOT. Specifically, before and after surgery, their performance was nearly perfect on filler trials where the target word was always consistent with the priming sentence and the non-target word was not in lexical competition with it (e.g., "This wall needs another coat of paint" has no /b/ word counterpart; pre-surgery: 100% correct performance in both participants; post-surgery: 98% for P1; 100% for P2). Their performance on catch trials where the target word was presented visually as a written word alongside a non-target word was also very high (pre-surgery: 94% for P1; 98% for P2; post-surgery: 92% for P1; 98% for P2). The overall behavioral results indicate that the surgical procedure exacerbated an impairment in the ability of both patients to adaptively integrate VOT information with semantic expectations from sentential context, amidst otherwise intact attentional or general meaning-related abilities, as assessed by the control tasks.

During the intracranial recordings before and after the ATL disconnection procedure in the operating room, the participants were awake and listened to a shortened passive version of the speech priming task. We ensured both patients were awake and engaged with the clinical and research teams during the two recording periods. Moreover, P2 took part in a simple tone detection task interleaved with the speech priming task to ensure that they were awake and engaged during the recordings. Their tone detection performance was high before (100% correct, 24/24 trials) and after (88% correct, 21/24 trials) the ATL disconnection procedure.

### Neuropsychological assessment results

Neuropsychological testing was conducted with the two patients, as shown in Supplementary Table 2. Reliable Change Index (RCI) results are presented in the table for all tests that the patients completed both before and after the surgery (see Methods). Post-operatively, both participants showed significant improvement in short-term memory for faces (WMS-III Faces I; RCI: P1: 3.402, P2: 2.91), which is not unexpected in medial temporal lobe (MTL) epilepsy patients[20]. No other tests showed significant changes consistently across both participants at the time of testing. Two language-related tests were administered to both participants, however, only pre-operatively: the Boston Naming Test (BNT) which is a test for naming drawings of objects, and the Controlled Oral Word Association (COWA) Test, which is a verbal fluency test. On the latter, none of the participants were impaired pre-operatively, and on the BNT, Participant 1 was right on the cusp of impairment (score of 48), while Participant 2 was in the average range (score of 55).

### Neural system state prior to ATL disconnection

We first studied the neural system state prior to ATL disconnection. Figure 2B shows the electrode coverage, which included left hemisphere IFG, superior temporal gyrus (STG) and Heschl's Gyrus (HG). Temporal pole contacts within the ATL were only viable pre-disconnection, therefore they were not used for any analyses that required comparison between pre- and post-disconnection.

Diffusion MRI data showed that the ATL in the two patients was structurally highly interconnected with frontal and temporal areas (Fig. 2C; Supplementary Fig. 2-3). The pre-surgical tractography analyses show the expected streamlines coursing between the ATL and prefrontal and more posterior temporal sites (Supplementary Fig. 2).

Node hubness can be quantified using graph theoretic metrics such as weighted degree centrality, defined as the total strength of links incident upon a specific node[51]. As a measure of hub centrality, we calculated the sum of each node's bidirectional (in and out) functional connectivity edge weights taken from the results of the effective connectivity analysis using state-space Conditional Granger Causality (CGC; see Methods). We evaluated the a priori hypothesis of the hub strength of temporal pole (TP) contacts relative to the other regions of interest (HG, STG, IFG pars opercularis and triangularis), and we found that in the gamma band the TP's hub strength is larger than the median hub strength of the other ROIs ($p = 0.036$, multiple comparisons FDR corrected).

In both participants, we observed the expected strong neurophysiological activity to the speech sounds in the auditory cortex (HG: Fig. 3B; STG: Supplementary Fig. 4), in the forms previously reported[52]. Moreover, the auditory cortex showed clear speech VOT representations, seen as systematically time-shifted auditory cortical potentials to the VOTs pre-disconnection (Fig. 3A, for exact peak times, see Supplementary Table 3). Speech prediction-related neurophysiological responses in HG were also observed, including mid-latency (270–350 ms) evoked responses after target word onset that were stronger for incongruent compared to congruent word conditions (Fig. 3C; Supplementary Fig. 5). The observed mismatch response components are similar to prior reports[43,49].

### Neural network alterations after ATL disconnection

We evaluated the neural network alterations post-disconnection by comparing the pre- and post-disconnection data. Figure 2A shows the post-surgical lesion site, determined from pre- and post-surgical structural T1-weighted images, which includes the ATL surgical procedure in both participants and the further MTL resection in P1. The cortical tissue affected only by the ATL disconnection procedure in both participants is shown in Supplementary Fig. 3.

We evaluated the structural connectome impact of the ATL disconnection procedure by comparing the diffusion MRI data obtained 2–6 weeks before the day of the procedure and removing all tracts that pass into resected area evaluated from the T1-weighted structural scans obtained 2 months after the surgical procedure. The analysis showed that the ATL disconnection affected interconnectivity with frontal and other temporal lobe areas via ventral pathways (Fig. 2C). Tracts particularly affected by the ATL disconnection post-surgery include the inferior longitudinal fasciculus, the uncinate fasciculus, and the cingulum bundle (Supplementary Fig. 2-3). Dorsal pathways between IFG and auditory cortex, including the arcuate fasciculus, were unaffected.

The surgical procedures and epilepsy pathologies were not identical in both participants, however, their overall connectivity matrices before disconnection were highly similar (Spearman's rho=0.82; Supplementary Fig. 3B). The key difference in the surgical procedures was that P1 had more extensive anterior temporal and medial temporal lobe removal than P2. Additionally, P1's ATL disconnection procedure also resulted in a reduction of tissue in the superior and middle temporal cortical areas, including some of the auditory cortical areas on the anterior STG (Supplementary Fig. 3A). Supplementary Table 4 shows that the patients' physiological parameters during the surgical procedure were comparable during the two recording periods. Anesthesia had been discontinued at least 20 min prior to both testing periods in both patients, with the ATL disconnection procedure completing ~23 min prior to the commencement of the intracranial recordings reported.

After ATL disconnection, we obtained evidence for both unaffected and altered site-specific neurophysiological signals, including functional interactions between sites. One of the few neurophysiological signatures that was minimally affected pre- and post-disconnection in both participants was the timing of the auditory cortical responses to the VOTs, which was unaffected in both patients for the initial VOT response Peak 1 (Fig. 3A; Wilcoxon signed-rank test respectively in Peak 1, P1: $Z = 4.0$, $p = 0.38$, P2: $Z = 1.0$, $p = 0.13$). The later Peak 2 response was not consistently different across the two patients (P1: $Z = 3.5$, $p = 0.38$, P2: $Z = 0$, $p = 0.031$).

In support of the modern diaschisis and predictive coding hypotheses (Fig. 1B), in HG we observed strikingly magnified post-disconnection speech-related high gamma responses that were significant in both participants (Fig. 3B; cluster-based permutation test, $p < 0.001$). With respect to lower frequency responses, beta band responses in HG during the target speech sound presentation were either enhanced or attenuated across the individual recording contacts (Fig. 3B). Theta and alpha band responses in HG after disconnection showed similar directionality of effects across the recording contacts as observed for beta. Higher-level stages in the auditory cortical hierarchy (STG) had largely reduced post-disconnection responses to the target word, in both participants, particularly for the gamma band (compare blue (pre) versus red (post) traces in Supplementary Fig. 4 for the STG). There was also evidence for significantly increased theta-based speech responses in the IFG pars opercularis in both participants (compare red (post) versus blue (pre) traces in Supplementary Fig. 4 for IFG op.).

Furthermore, consistent with the predictive coding hypothesis was the observation that the neurophysiological semantic mismatch response was diminished post-disconnection in HG (Fig. 3C; cluster-

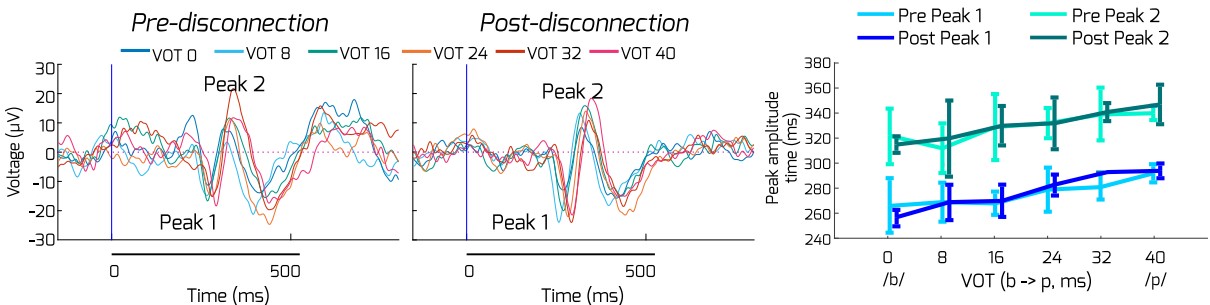

**(A) Speech VOT responses largely intact post-disconnection**

**(B) ATL disconnection speech representation impact**

**(C) Congruent vs. incongruent speech prediction effects**

**(D) Combined in- and outflow**

based permutation test, $p < 0.05$). The pre-disconnection congruency mismatch response is seen in Fig. 3C as a difference wave (congruent versus incongruent; blue line) shows the expected larger congruent/incongruent responses throughout the target word, similar to previous reports[49]. The post-disconnection response is significantly reduced in both patients (gray bars in Fig. 3C), seen as post-disconnection mismatch responses close to 0 difference (red line in Fig. 3C). The

presented results are robust to the choice of baseline correction period (Supplementary Fig. 6).

To identify whether and which post-disconnection effects resulted in altered directional functional connectivity, we relied on the spectrally resolved state-space CGC analysis[53] combined with graph theoretic analysis of hub centrality as a measure of network changes of effective connectivity across the frequency bands. The CGC effects

**Fig. 3 | ATL disconnection impact on auditory cortical speech responses.**
**A** Heschl's gyrus (HG) responses to the different VOTs in the speech sounds, averaged across the two participants and recording contacts ($N = 8$, 4 in each participant). The onset of the target word is 0 on the horizontal axis. Peak 1 depicts the negative peak around 250 ms post-stimulus onset, Peak 2 depicts the positive peak around 350 ms post-stimulus. Right: mean VOT peak amplitudes (error bars are +/- standard deviation (SD) across participants) showing the linear relationship between peak times and the different VOTs for the average waveforms.
**B** Neurophysiological responses to the target words are selectively altered in HG. Left plots show the event-related spectral perturbations (ERSPs in dB) across the frequency bands (theta, alpha, beta, low and high gamma). The right panel shows a summary for beta (13–30 Hz) and high gamma (70–150 Hz; Supplementary Fig. 4 shows the other frequency bands). Top plots show speech responses averaged across contacts and the two participants. Bottom plots show the consistency of effects by participant and exemplar contacts. Blue lines show the pre-

disconnection and red lines the post-disconnection responses (error bars are +/- standard error of mean (SEM) across trials). Statistically significant differences between pre-and post-disconnection are shown in green bars (cluster-based permutation test, two-sided, $p < 0.001$) underneath the line plots. **C** Difference plots of the auditory evoked response in HG to the speech sounds, contrasted by whether they were preceded by a congruent or incongruent biasing sentence. The blue line shows the pre-disconnection congruency vs. incongruency mismatch response (MMR), red line shows post-disconnection MMR, the shading shows the SEM across trials. Statistically significant differences between pre-and post-disconnection are shown in gray bars (cluster-based permutation test, two-sided, $p < 0.05$ for at least 25 ms) underneath the line plots. **D** Conditional Granger Causality measure of hubness showing the TP is the most functionally hub-like in the gamma band compared to the other ROIs ($p = 0.036$, FDR corrected). Larger node size depicts larger hubness value. See Supplementary Table 6 and Supplementary Fig. 7.

that significantly differed post-disconnection are shown combined and by participant in Fig. 4. Two observations are most striking. First, although there was a general disruption in effective connectivity, seen as significant decreases in CGC values post-disconnection (purple arrows, Fig. 4), certain interactions between the frontal and auditory cortical sites increased post-disconnection (green arrows, Fig. 4). Consistently in both participants, the CGC measure revealed increases in effective connectivity in the nodes across the recorded fronto-temporal network, particularly in the gamma band (green bars in Fig. 4A; larger dots at each node in Fig. 4B). These connectivity effects cannot be easily explained by gamma power changes (Supplementary Fig. 4), which were only significant in HG (Fig. 3B). The increased gamma-based effective connectivity seems to indicate an inter-connectivity impact on the broader network after the loss of the ATL. Second, relative decreases in centrality dominated the network in the lower frequency bands in both participants (particularly in theta; purple bars and dots, Fig. 4), showing more focal interactions in these lower frequency bands.

Although we anticipated increases in functional connectivity being driven by prefrontal cortex[34,35], other areas can also drive compensatory responses[27,32,33]. The spatial pattern of increased effective connectivity that we observed post-disconnection appeared to be either frontal or temporal lobe led across the two participants (pattern of green arrows, Fig. 4B). Specifically, P1, whose ATL disconnection procedure affected anterior superior temporal lobe areas (Supplementary Fig. 3A), exhibited magnified interconnectivity that appears to be driven by the IFG, as expected (Fig. 4B). P2, who had intact superior temporal lobe areas following ATL disconnection (Supplementary Fig. 3A), exhibited magnified functional connectivity that appears to be driven by the temporal areas (Fig. 4B). Altogether, the effective connectivity results suggest that both frontal and temporal lobe areas could be involved in driving the observed increases in effective connectivity.

## Discussion

We report rare intracranial recordings in two neurosurgery patients obtained minutes before and after a neurosurgical treatment that required disconnection of the left ATL. The results cannot support the 'no disruption' or 'solely disruption' (classical diaschisis[30]) hypotheses after the loss of the ATL. Instead, our results provide support for the incomplete immediate compensation hypothesis, largely in the form of neurophysiological impact and alterations stipulated by the predictive coding framework and more generally by a modern diaschisis hypothesis[31]. The study results thus show direct evidence for the important role of the left ATL in semantic processes and predictions, as they also indicate remarkably rapid alterations by the distributed network nodes, observed largely as anticipated by the predictive coding framework. Given that these are rare recordings in two patients with the requisite similarity in fronto-temporal recording coverage,

this is by necessity a case-of-cases study that required a within-subjects design and significance testing of pre- and post-surgical effects. As such, the results are shown both individually and combined, and we only interpret results that are robust, significant, and consistent in both participants, as follows.

In both participants, removal of the ATL resulted in a robust impact on and alteration in neural signaling between IFG language sites and temporal lobe auditory speech processing sites, distant from the surgical site. Before disconnection, the ATL showed hub-like structural and functional connectivity with the other nodes, consistent with its proposed centrality in the hub-and-spokes model[4,26]. Evoked responses to physical acoustical features, namely voice onset time[54,55], were minimally affected by ATL disconnection, while speech prediction signals and processes were altered in remarkably specific ways, as follows.

The predictive coding framework[36,38] anticipated both the form of alterations and possible compensation in the case of a loss of ATL prediction-related signals, provided that the remaining brain areas would not be able to immediately fully compensate for the loss of the ATL. First, the framework predicted the magnified gamma band responses to speech sounds in auditory cortex (Fig. 3B). Second, the framework predicted the diminished speech-prediction related (mismatch) response (Fig. 3C), the features of which in the observed pre-disconnection responses are similar to auditory mismatch responses in prior reports[36,38]. Third, we observed increased effective connectivity across the recorded nodes (CGC, Fig. 4A) in the gamma band, suggesting that the gamma responses to the target speech sounds were strongly affecting gamma-based interconnectivity across the network after the loss of the ATL. The depletion of the speech mismatch response together with the magnification of the high gamma speech response in HG after the loss of the ATL suggests that prediction error signaling is abnormally magnified, potentially causing a drastic reduction in the dynamic range of perceptual inference signals interacting with sensory processes in auditory cortex.

There was evidence of attempted incomplete compensation in the forms stipulated by both the predictive coding[34,35] and modern diaschisis frameworks[31]. This included more focal (Fig. 4) and increased effective connectivity within and between the frontal and temporal lobe sites. However, the intact fronto-temporal sites capable of compensating for a loss of speech-related and expectancy signaling in the neural network did not appear to be able to recover the diminished neurophysiological speech incongruency responses in HG after ATL disconnection in this acute phase (Fig. 3C). In addition, we observed a lasting pre- versus post-operative behavioral impact on the two patients' abilities to integrate speech semantic and perceptual information two months after the surgery (Fig. 1E), although we cannot rule out that the additional MTL resection step in P1 may have contributed to their behavioral changes. Moreover, inter-regional theta-based synchronization is often associated with effective encoding and retrieval in memory tasks[32]. Interestingly here, theta-based effective

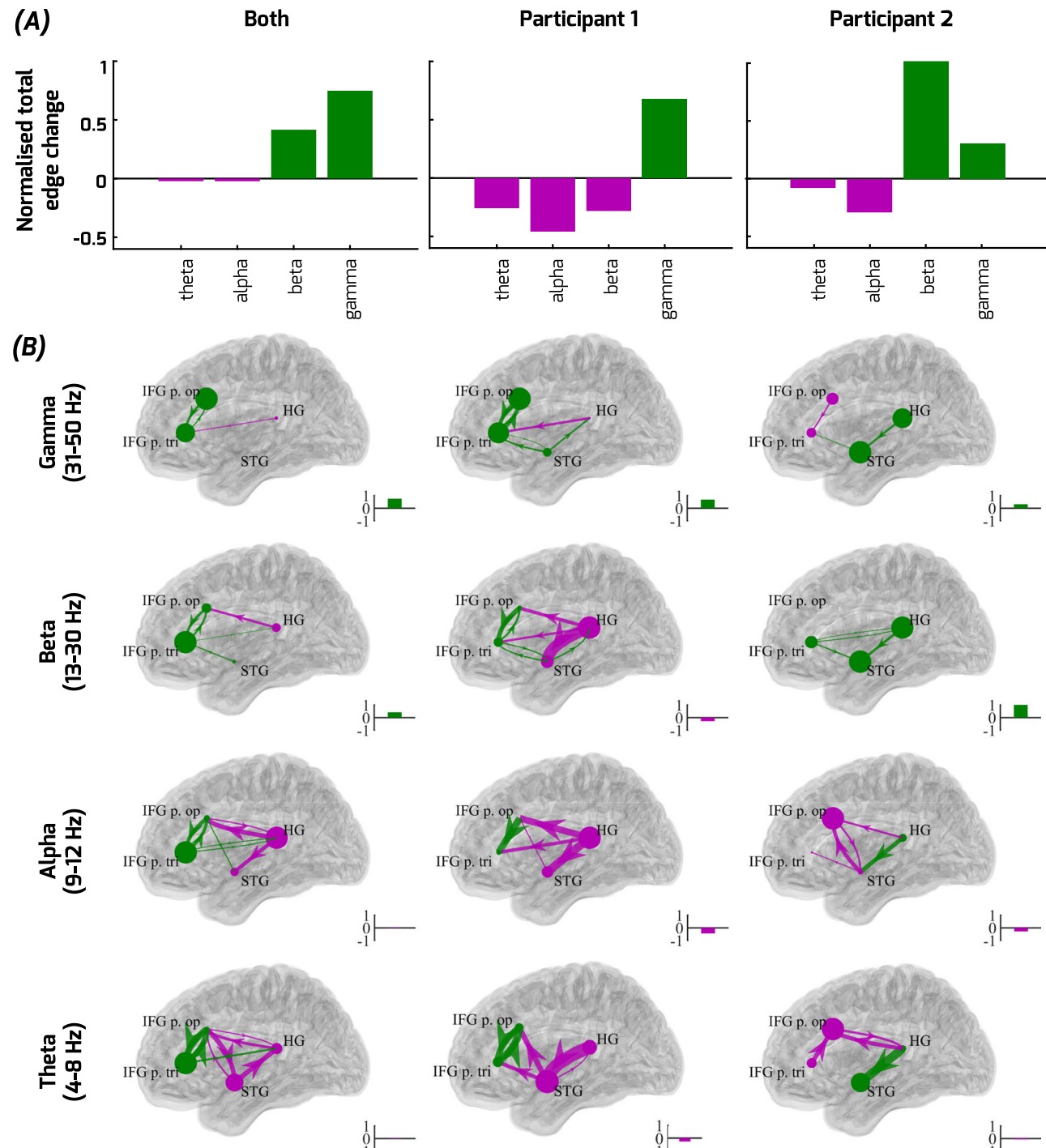

**Fig. 4 | Effective CGC connectivity alterations combined and by participant.**
Significant increases (green) or decreases (purple) post-disconnection in CGC
effective connectivity are shown summarized as normalized total network edge
weights (**A**) and overlaid on a template brain (**B**), based on Supplementary Fig. 8.
Results are shown for both participants combined and each participant separately
(graphs, left to right), further separated by frequency bands. **A**, **B** For each plotted
graph, total network increases/decreases in edge weight are shown normalized
relative to the total significant absolute edge weight changes (inset bars; also shown
aggregated into summary plots, bottom). **B** shows edge weights (line thickness)
and node hubness (inflow+outflow; node size/color) depicted overlaid on a
template brain. Note how the normalized total network edge weight changes reveal
that both participants exhibit relative increases (green) in network-wide functional
connectivity in the gamma band, and relative decreases (purple) in lower frequency
bands. Only FDR corrected directional edges showing significant differences
between pre- and post-disconnection based on permutation tests are shown in
panel B. Despite these consistencies across both patients, the spatial pattern of
increased functional connectivity post-disconnection differed across the frontal
and temporal lobes in the two participants increases in effective connectivity, see
manuscript text for details.

connectivity was disrupted after ATL disconnection, further underscoring the importance of the ATL as a brain hub for speech prediction and processing.

The 'no impact' (complete compensation) hypothesis can be ruled out by the results because there was substantial neurophysiological disruption after the loss of the ATL. The results can also rule out a purely disruption (classical diaschisis[30]) hypothesis, where disconnection of the ATL would result solely in uncoordinated disruption to intact, pre-surgically interconnected sites. The results were more consistent with neurophysiological alterations largely in the forms anticipated by the predictive coding framework hypotheses, in specificity, and by the modern diaschisis hypothesis, more generally. The results provide direct evidence in support of the hub-and-spokes model, in the forms stipulated by a modern diaschisis and predictive coding framework, indicating that the left ATL has a vital role in speech processing and expectancy that is not fully recoverable in this acute phase. An incomplete functional compensation interpretation fits within established theoretical frameworks where, for example, functional compensation does not result in fully stabilized neurobiological network signals or behavior[34], as we see here. The IFG, with its established role in semantic and syntactic language functions[8,9,26] is a key candidate region to coordinate compensation after the loss of the left ATL, together with the right ATL and other auditory and speech processing sites. However, the intact areas, including the IFG and right ATL, not directly physically affected by the ATL disconnection procedure, were unable to completely recover from the loss of the ATL during this acute phase. Nonetheless, these other sites did show evidence for remarkably rapid and specific alterations, underscoring the importance of the distributed network nodes and other hubs[5,7,15].

The neurophysiological results therefore broadly support predictive coding and related frameworks[56,57], with testable extensions. The frameworks can now be applied with greater confidence to model acute neural system impact and alterations in physical and artificial neural networks, informed by neurophysiological data recorded directly from the human brain. An intriguing observation in terms of the predictive coding framework was how consistently the alterations in interconnectivity between frontal and auditory sites also involved the theta frequency band. The role and importance of theta oscillations in the predictive coding framework is growing[58], having been associated with both feedforward (entrainment to the speech syllabic rate[59,60], cognitive sampling at a theta rhythm[61,62]) and feedback processes[63]. Theta-based interconnectivity is also a prominent feature associated with cognitive function[27,32,33].

Although we were only able to study two patients here, there are several reasons to believe that these results are generalizable to other patients with epilepsy, and to individuals with healthy brains. Firstly, they are consistent across individuals despite significantly different epilepsy etiologies and timescales. P1 had hippocampal sclerosis and secondary gliosis of the temporal lobe, an acquired and progressive condition, while P2 had a congenital cavernoma present since birth. Secondly, our patients' neurophysiological system shows fairly normal structural connectivity pre-operatively, including speech-related and mismatch responses in HG and the expected pre-operative speech-related responses in IFG (Fig. 3B). Thirdly, previous literature comparing the semantic networks of patients with chronic anterior temporal lobe epilepsy to healthy individuals using neuroimaging (Fig. 1 in ref. 64) shows that the area of maximal semantic activation in patients and controls is in the same locus, despite this overlapping with the epilepsy focus in the patients. Finally, we are making only comparisons between pre-resection and post-resection in the same individuals, rather than making any comparisons to connectivity in controls that might be confounded by pre-surgical connectivity changes as a result of epilepsy. Thereby, the overall findings suggest that the architecture of, impact on and forms of attempts at immediate compensation in these networks is well conserved despite pathology.

In summary, this study provides insights into the language connectome immediately before and after neurosurgical treatment affecting the left ATL. The results underscore the limitations of cognitive models that either stipulate an irrecoverable functional role of the ATL or the possibility of complete immediate neurophysiological compensation by the rest of the network. The findings indicate that the left ATL is likely to act in two key ways: 1) as a key node in a constructive network for semantic processing and predictions involving auditory cortex; and 2) in interaction with language-related areas in prefrontal cortex, which have a proposed role in semantic control[65].

Given the rarity of ATL disconnection procedures, and changes in epilepsy patient monitoring away from grid recordings and treatment procedures towards neuromodulation, such datasets will likely become even more scarce over time. Consequently, it is important for future studies to establish direct correspondence to source-localized EEG[56,57,66] or fMRI data[67,68] in the same patients. Although fMRI has reduced temporal precision and EEG has lower spatial specificity compared to intracranial EEG, the combination of fMRI and EEG can together form a coherent noninvasive neuroimaging picture on neural system impact and compensation when validated with intracranial recordings in the same patients. Furthermore, validated non-invasive imaging biomarkers obtained pre- and post-operatively would allow testing the patients throughout the recovery period to advance insights into neural network impact and compensation in larger patient cohorts.

We speculate that the form of rapid alterations seen here may engage early gene expression and adaptive brain plasticity mechanisms known to occur in tens of minutes[69]. The observed site-specific alterations in these two patients provide a foundation for human or animal model research to further understand the cascade of neurophysiological reorganization and compensatory mechanisms in neural networks following neurological impact and treatment.

## Methods

### Participants and ethical approvals

The research protocols were approved by the Institutional Review Board at the University of Iowa (IRB #200112047). Written informed consent was obtained from both participants. Participation in the research protocols was voluntary and did not interfere with the participants' clinical treatment. The participants had the right to revoke consent at any time without affecting their clinical evaluation and treatment. Participant recruitment and research were conducted in accordance with the latest ethical principles and practices for conducting invasive intracranial research experiments in human patients[70]. This included maintaining the integrity of clinical care and space and ensuring the voluntariness of participation in research protocols at every stage. The research was carried out in one of the operating rooms at the University of Iowa during surgery for electrode removal and seizure focus resection.

### ATL neurosurgical disconnection procedure

We recorded data from the left hemispheres of 2 neurosurgical patients (P1: 63-year-old woman, P2: 32-year-old man). Both were native English speakers, right-handed and left hemisphere language dominant based on the Wada test. They had been diagnosed with medically refractory epilepsy and were undergoing surgery to remove seizure foci during acute electrocorticography (ECoG) monitoring in the operating room. The surgery required the participants to be awake and able to interact with the clinical team for at least two periods of clinical motor speech and naming area mapping using electrical stimulation in order to minimize post-surgical morbidity[71] (Supplementary Table 4).

An ATL disconnection procedure was performed for neurosurgical treatment, whereby to access the surgical treatment site a portion of ATL tissue is left intact within the middle cranial fossa and remains

functionally disconnected from the remainder of the temporal cortex (transecting both gray and white matter)[72–74]. P1 had epilepsy caused by hippocampal sclerosis and secondary gliosis of temporal lobe. They participated in a two-step surgical procedure starting with the disconnection of the ATL. This made it possible for the clinical team to access the deeper epileptogenic foci structures for resection involving the hippocampus and parts of the amygdala, the second step in P1's procedure (Supplementary Fig. 3 shows the tissue removed during both steps). For P2, the procedure only required the first step, disconnection of the ATL to resect the epileptogenic cavernoma located in the temporal pole (Fig. 1B; Supplementary Fig. 3B). For both P1 and P2, the intracranial recordings involve solely the pre- and postdisconnection stages and the tissue removed after only the ATL disconnection stage (Supplementary Fig. 2). No further recordings as part of this study were made or were possible after the additional MTL resection step in P1. For both participants, we recorded data as they were awake before and after the ATL disconnection. The recording contact coverage is shown in Fig. 1C. This included temporal pole contacts which only provided viable recordings before the ATL disconnection (Supplementary Fig. 3).

## Speech stimuli and behavioral task

The semantic prediction task, tested approximately a month before and 2 months after the surgical procedures, involved listening to sentences that primed the perception of a final target word (Fig. 1E), based on[48]. It included naturally produced sentences to semantically bias the listener to expect a target word, beginning with either a /b/ or /p/ phoneme, at the final position in the sentence. The target word was manipulated along a VOT continuum with 6 steps (0 ms to 40 ms) to move evenly from the perception of /b/ as in 'bark' (short VOTs) to /p/ as in 'park' (long VOTs). Both endpoints corresponded to a real word, creating naturalistic conditions where the biasing sentence and the target word corresponded to the listener's expectation. Expectations were (congruent condition) or were not met (incongruent condition). Sentence primes were counterbalanced with target words from each step of the VOT continuum (Fig. 1E).

During the perioperative behavioral testing, the participants performed an extended version of the task. They were seated in an electrically shielded, sound attenuating room. During the semantic predictions task, 7 different word pairs were used (bill/pill, bad/pad, bath/path, back/pack, bark/park, bowl/pole, beach/peach), amounting to 504 experimental trials, not including 126 filler and 63 catch trials. The filler trials were sentences that fully predicted the target words (e.g., "The walls need another coat of paint", where paint is the target word which has no /b/ counterpart). The catch trials consisted of sentences, where after hearing the sentence, the target word was presented visually on a screen along with another written word to choose from. These could either be fully predictive sentences as in the case of the filler trials, or fully congruent sentences taken from the main experiment. The participants were asked to press a button after each sentence to indicate the starting phoneme (b or p) that they heard and perceived in the target word, or the word that best fit the sentence in the case of the catch trials (e.g., paint vs pros). Altogether, the entire semantic predictions task and filler/catch trials consisted of 693 trials and took about 75 min to complete. For a more detailed description of the experimental stimuli, see Supplementary Table 5. Before the surgery, this task was also used to determine which 4 sentences had the biggest effect to focus on during the shorter data collection period possible in the operating room. There are published prediction trials demonstrating that lifelong learned associations and semantic priming cannot be easily overridden by immediate experimental context (e.g.,[12,43]), and mismatch responses may be attenuated in a context where incongruence is frequently occurring but are difficult to abolish[75].

The phoneme identification responses on the semantic prediction task were statistically tested using a logistic mixed effects model with the three within-participant factors: VOT (6 levels: 1 to 6, centered), Semantic bias (2 levels: /b/ and /p/ bias, +/−0.5), and Disconnection (2 levels: before, and 2 months after the surgical procedure, +/−0.5). These were each entered as main effects along with the two-way interactions with disconnection. As these analyses were conducted individually for each subject, the only random factor was Word pair (7 pairs). Inferences allow us to generalize to the subject's own behavior, not the population. Following[76], we used the maximal random effects structure for both participants. Covariance terms among the random slopes were dropped to facilitate convergence. This led to the model given in (1), stated in R's lmer() notation (see Supplementary Table 1 for complete results):

$$P\ responses \sim (VOT + Bias) * Disconnection + ((BiasP + VOT) * Disconnection || Word\ pair) \tag{1}$$

During the recordings in the operating room, the participants performed a passive version of the semantic predictions task (not required to press buttons to the target words) to elicit neurophysiological activity with the stimuli and conditions that they experienced during the pre-and post-operative testing. Stimulus delivery was controlled by the Psychophysics Toolbox[77,78] running in MATLAB 2018b (The MathWorks, Inc., Natick, MA). Four different word pairs were used with three different preceding sentences (12 different sentences altogether) that biased listeners to expect one endpoint (/b/ or /p/) over the other, totaling 288 experimental trials. An additional 72 filler trials were also included, consisting of sentences that fully predicted their target word, to ensure that most trials fulfilled semantic expectations. Stimuli were presented at a comfortable level through a speaker placed in front of the participants. Each recording session was divided into 4 blocks so that the experimenter could ensure the participant was awake and engaged with the task. For P2, to additionally ensure the participant was awake and attending to the sounds presented, they participated in a simple tone-detection task during the speech priming paradigm. They were asked to press a button they were holding in their dominant hand whenever a 440 Hz, 250 ms long sine wave auditory tone was presented (24 randomly occurring tone detection trials in between the speech priming task). The first recording session was done after the clinical recording electrodes were placed, but before the ATL disconnection procedure, completing in 25–30 min. After the ATL disconnection procedure, a second similarly timed recording session with the same task ensued.

## Neuropsychological testing

As part of pre-operative clinical workup, both participants were evaluated using a standard neuropsychological battery. P1 was assessed about 9 months before surgery and P2 about 16 months before. To assess change in cognition over time for research purposes, participants were reassessed post-operatively during the chronic period of recovery (>3 months). P1 was seen 7.4 months following resection, and P2 was seen at both 6.6 and 14.2 months. Due to time constraints, not all neuropsychological measures were possible to be re-administered, and P2's second research visit allowed for more testing than P1. Tests administered and their results are reported in Supplemental Table 2. Pre- and post-operatively, participants were assessed using the Rey Auditory Verbal Learning Test (RAVLT)[79], select subtests from the Wechsler Adult Intelligence Scale 4th Edition (WAIS-IV)[80], select subtests from the Wechsler Memory Scale 3rd Edition (WMS-III)[81], Trail Making Test (TMT)[82], Lafayette Grooved Pegboard Test (GPT)[83], Boston Naming Test (BNT)[84], Controlled Oral Word Association Test (COWA)[85], Beck Depression Inventory-II (BDI-II)[86], and Beck Anxiety Inventory (BAI)[87]. For the purposes of comparing participants' preoperative performance to population norms, z-scores were calculated using age- and education-matched means and standard deviations available from test manuals. To assess change in performance

following ATL disconnection, we calculated a reliable change index (RCI) for each test. This index accounts for test-retest reliability to better predict meaningful change over time[88,89]. We used population standard deviation and test-retest reliability values from work done on populations with epilepsy[90–92]. RCI scores of ±1.96 are considered significantly altered between timepoints, suggesting meaningful improvement or decline.

### Electrode coverage and neurophysiological data collection

Clinical platinum-iridium electrodes (AdTech, Racine, WI) were used to monitor interictal activity. The plans for placing ECoG electrodes for acute intraoperative clinical monitoring were formulated during a preoperative multidisciplinary epilepsy management conference. Four contact electrode strips were placed on the temporal pole, under the temporal lobe, and over the lateral surface of the superior temporal lobe. An 8-contact depth electrode was placed in HG using Stealth navigation (Medtronic). Two subgaleal electrodes were used as ground and reference. A 64- or 96-contact electrode grid (P2 and P1, respectively) was placed over the frontal lobe, containing reference and ground contacts. All contacts except the frontally placed grid remained in place for the entire recording time. The location of the frontal grid post-disconnection was checked and any displacement of recording contacts in relation to their pre-disconnection location was corrected. The positions of the frontal grids, and the STG electrode were confirmed after the surgery with the help of intraoperative photographs or using ultrasound[93] in the case of HG to reconstruct the recording channels' anatomical locations. The temporal pole and the subtemporal area locations could not be confirmed this way because they were occluded by the skull and are thus only approximated in location (Fig. 2B). The safety and clinical treatment utility of having the electrode coverage used here, including supratemporal HG electrodes has been previously demonstrated[94,95]. For instance, the pattern of ictal spread of seizure activity to the supratemporal plane is highly predictive of subsequent seizure-free outcome rates, and when high levels of interictal activity across the HG supratemporal plane electrode channels are observed, this is considered in the clinical resection plan.

### Structural and diffusion-weighted MRI

About a month before and approximately 2 months after the surgery, we obtained T1-weighted structural MRI scans (fast spoiled gradient-echo sequence, 1 x 1x 0.8 mm resolution, TR = 8.508, TE = 3.288, flip angle = 12°, 32 slices) and a 64-direction diffusion-weighted MRI (1 x 1 x 2 mm resolution, TR = 10332, TE = 88.9, flip angle = 90°) using a Siemens 3T Discovery MR750w scanner. Network construction methods broadly follow those described previously[56,96]. Preoperative T1-weighted MRI scans were processed using FreeSurfer version 6.0 using the recon-all tool to generate cortical regions-of-interest (ROI)[97]. The default parcellation scheme from FreeSurfer (the Desikan–Killiany atlas[97,98]) was used, which contains 82 anatomical cortical and subcortical ROIs (e.g.,[99,100]). These regions were also combined to generate a gray matter ribbon using FSLmaths (later used as a seed region for surface-based tractography). Masks covering resected tissue were generated by first linearly registering (6 degrees of freedom) the postoperative T1-weighted MRI to the pre-operative T1-weighted MRI using FSL FLIRT. Then, using FSLview, we manually defined the resected tissue in pre-operative T1w space by overlaying the scans. Care was taken to avoid areas known not to be resected (see ref. 96). Supplementary Fig. 3A shows the areas impacted by the disconnection of the left ATL in both participants.

Susceptibility distortions on preoperative diffusion-weighted MRI data were corrected using the SynB0 DisCo tool in conjunction with the FSL TOPUP tool[101,102]. The FSL tool 'EDDY' was used to correct for motion and eddy current distortions with the bvecs rotated appropriately[103]. The diffusion data were then reconstructed using generalized q-sampling imaging with a diffusion sampling length ratio of 1.25[104]. A deterministic fiber tracking algorithm was then used[105]. Tractography seeds were placed in the gray matter ribbon and deep brain structures by registering the preoperative aparc+aseg file generated by FreeSurfer to the preoperative diffusion scan (rigid body using orig.mgz DSI Studio). Tractography parameters were as follows: The anisotropy threshold was randomly selected. The angular threshold was 60 degrees. The step size was randomly selected from 0.5 voxel to 1.5 voxels. The fiber trajectories were smoothed by averaging the propagation direction with a percentage of the previous direction. The percentage was randomly selected from 0% to 95%. Tracks with lengths shorter than 15 mm or longer than 150 mm were discarded. A total of 5,000,000 tracts were calculated. Preoperative structural connectivity networks were then calculated as the number of streamlines ending between each region pair. Diffusion MRI tractography streamlines are quantified and interpreted as representing white matter connectivity. To compute expected postoperative networks, we removed all tracts which pass into the mask voxels and recomputed the network. The difference between the two networks shows the percentage change in connectivity which we calculated as post- vs pre-disconnection. Networks were visualized using brainnetviewer[106] (Fig. 2C).

### Intracranial ECoG acquisition and analysis

The ECoG data were sampled at 2000 Hz and filtered (0.7–800 Hz bandpass, 12 dB/octave roll off) using Tucker Davis Technologies system (TDT, Alachua, FL), then downsampled to 1000 Hz, denoised using demodulated band transform (DBT[107]; bandwidth = 0.25). Singular value decomposition was used to discard the first principal component based on the covariance matrix calculated from the highpass filtered (300 Hz) data. Event-related spectral perturbations were calculated with EEGlab[108], epoched around the target word onset by log-transforming the power for each frequency and normalizing relative to the baseline (mean power in the pre-stimulus reference silent interval −150 to 0 ms before target word onset). The frequency band-related activity was averaged across all trials. The responses measured in each of the frequency bands (theta, alpha, beta, low and high gamma) were statistically compared using cluster-based non-parametric permutation testing, as described in[109], testing the contrast between the pre-and post-disconnection states (10,000 permutations, $p < 0.05$). We excluded any significant clusters shorter than 25 ms in duration.

Event-related potentials (ERPs) were calculated with respect to the semantic prediction context (i.e., whether the word was within a predicted congruent sentence context or an incongruent context). This allowed us to evaluate prediction mismatch-related neurophysiological responses (−150 to 800 ms from stimulus onset; baseline corrected and filtered off-line using a 0.5 to 50 Hz band-pass finite impulse response filter; Kaiser–windowed, Kaiser β = 5.65, filter length = 3624 points). We calculated the ERPs to the congruent predicted condition using the 0 ms and 8 ms VOT target words for the /b/ bias sentences, and 32 ms and 40 ms VOT target words for the /p/ bias sentences. These were contrasted with the incongruent condition (e.g., prediction error) when the sentence created a bias towards expecting a /p/ word and the target word started with a /b/, using the 32 ms and 40 ms VOT target words for /b/ bias sentences, and the 0 ms and 8 ms VOT target words for /p/ bias sentences. The comparison included 96 trials each, and we subtracted the incongruent from the congruent ERPs to obtain a difference wave. To account for the VOT differences expected on the neurophysiological response signals (Fig. 2A), we matched the congruent vs. incongruent conditions by VOT, extracting incongruent target words with the shortest VOT from congruent words with the shortest VOT, etc. We then compared the difference between the recordings before and after ATL disconnection. The significant difference between the pre- versus post-disconnection speech response was evaluated using cluster-based permutation testing.

All the available channels in each participant and ROI that had a significant speech response in the pre-disconnection data were evaluated and used for further analysis. Significant speech responsiveness was calculated by comparing the baseline pre-stimulus neurophysiological activity with the activity during the target word. If the latter exceeded 2 standard deviations from the baseline variability for over 125 ms of the 500 ms target word length, the channel was identified as having a significant speech response. This procedure identified 4 channels for each participant in HG and 3 in STG. The HG and STG electrodes remained in the same location pre- and post-operatively. Moreover, all of the speech-responsive contacts identified pre-disconnection remained significantly speech-responsive post-disconnection, with only one additional recording contact in P2 in the STG showing a significant speech-responsive after disconnection that was not significant pre-disconnection. For the TP contacts, two contacts were averaged in each subject that showed significant speech response pre-disconnection. After disconnection the tissue in the TP was no longer viable, therefore TP contacts were not included in the analyses post-disconnection. For the IFG recording the grid required removal by the neurosurgeon before the ATL disconnection procedure and was replaced in a similar location for the post-disconnection recordings. We used a nearest neighbor co-registration approach based on pre-operative T1-weighted imaging and intra-operative photographs of the electrode locations, providing the coordinates of electrode locations both pre- and post-disconnection. The Euclidian distance of IFG grid displacement was on average 8.73 mm (SD: 2.03) for P1 and 4.77 mm (SD: 2.07) for P2. The IFG channels were included in the analyses only if they were localized to the IFG pars opercularis or triangularis after correction for the location of the IFG recording grid post-disconnection. Fewer contacts in the IFG were available post-operatively because the post-operative placement of the IFG grid was somewhat more posterior in both participants. IFG recording contacts in these regions were also only included for further analysis if they showed a significant speech response in the pre-operative data. For P1, this included 24 recording contact, respectively pre- and post-disconnection, for the pars triangularis, and 11 and 10 channels, respectively pre- and post-disconnection, for the pars opercularis. For P2, IFG recordings from pars triangularis included 11 and 9 channels, respectively pre- and post-disconnection, and 8 and 5, respectively pre- and post-disconnection, for the pars opercularis, respectively pre- and post-disconnection. The average displacement of the IFG channels pre- versus post-disconnection with significant speech responses was 1.71 mm (SD: 0.91) for P1, and 3.28 mm (SD: 1.35) for P2.

## State-space conditional Granger causality

Spectrally resolved state-space Conditional Granger Causality (CGC) analysis was used to investigate the directional neurophysiological interactions between brain regions involved in the classification of the target speech word. The method is multivariate and conditional in the sense that simultaneous time series from a collection of electrodes are included to account for direct and indirect influences between contacts. ECoG recordings were downsampled to 100 Hz and sectioned into trials, 0 to 2 s relative to the presentation of the target word. Intuitively, CGC tests whether activity in a source area can predict subsequent activity on a target area better than the target area can predict directed activity, as should be the case if the source area modulates activity in a recipient area. Prior to spectral CGC analysis, the mean at individual time points across trials was subtracted from the single trial ECoG data and then scaled by the standard deviation[53]. CGC considers the predictive effect of all other contacts which allows us to distinguish between direct influences of interest and indirect influences.

The state-space approach aims to address several significant problems in applying standard vector autoregressive (VAR) based CGC models to ECoG recordings, which are related to downsampling and nonstationarities[110]. The state-space model also addresses a number of theoretical and practical problems related to spectral CGC estimation[111–114]. Spectral CGC was directly computed using Geweke's[115,116] formulations based on the estimated state-space innovations covariance matrix, cross-spectral densities, and transfer functions[111,113]. State variables can be reconstructed from the measured ECoG recordings but are not themselves measured during an experiment. For modeling directional influence in the brain, it is possible to directly express the interactions between different regional signal time series as a state-space model. The state-space model in innovations form is defined by:

$$x_{t+1} = Ax_t + K\varepsilon_t \qquad \text{state transition equation} \qquad (2)$$

$$y_t = Cx_t + \varepsilon_t \qquad \text{observation equation} \qquad (3)$$

where $x_t$ is an unobserved (latent) m-dimensional state vector, and $\varepsilon_t$ is the vector of the innovations or prediction errors. The observed vector of time-series $y_t$ corresponds to the ECoG recordings from regions in the targeted network. The state transition matrix $A$, observation matrix $C$ and the steady-state Kalman gain matrix $K$ are estimated using a subspace method. Subspace methods are optimal for state-space model parameter estimation, especially for high-order multivariable systems[117]. The order of the state-space model was determined by the number of principal angles that differ from $\pi/2$ when estimating the data-driven state-space model by the subspace method[118]. Principal angles have been proven to be related to the singular values in singular value decomposition[118]. The order of the state-space model was 15, which corresponds to the vector size $x_t$. Time series recorded from individual contacts were clustered together by averaging multiple simultaneous recordings to form ROIs. Post-stimulus time series from the two participants were concatenated as trials in the combined data.

The state-space variant of Granger Causality was used because of the necessity for assessing directional effects conditional on the other recorded nodes and reference electrodes and its capability to deal with indirect, including volume conduction, artifacts. The approach is conditional in the sense that simultaneous time series from a collection of electrodes are included to account for direct and indirect influences between electrodes. Although accounting for all possible indirect sources is not possible if those are not directly recorded, the intent is to account for a subset of indirect sources and effects upon them by conditioning. There is a concern that indirect sources and volume conduction could induce instantaneous correlated noise across recording electrodes with a lag that appears like effective connectivity. To minimize this concern, we routinely quality control the data by, for example, re-referencing to ground or other contacts. Also, the state-space method of calculating CGC has been shown to be more robust than other methods to the effects of volume conduction common to EEG analysis. For instance, simulation studies[119] show that state-space CGC[111] is able to model volume conduction and the evolution of the neural state within the same framework. These studies show that the state-space approach can result in the correct recovery of the original causal structure of the dynamical system in the presence of volume conducting noise. Cheung et al.[120] also shows that CGC analysis using state-space models was less sensitive to observation noise compared with two-stage VAR-based models. Faes and colleagues'[112] show that the state-space method yielded highly accurate spectral estimates of CGC that followed expected profiles over coupled directions and negligible magnitudes over uncoupled nodes. The state-space CGC estimator also showed higher reliability compared with standard VAR-based methods.

To statistically evaluate the reliability of the connectivity results we used a phase-randomization surrogate data technique to construct an empirical null distribution[121–123] representing chance influence

 

between ROIs. This method consists of randomly shuffling the Fourier phases of each of the ECoG recordings which generates uncorrelated data with preserved autocorrelation properties. The matrix of spectral CGC between four ROIs in HG, STG and IFG opercularis and triangularis was statistically evaluated as follows: 2000 surrogates were generated and thresholded at $\alpha = 0.05$ of the null distribution, values below which were subtracted from the observed spectral CGC and trimmed at zero. Finally, statistically significant pre- and post-disconnection spectra were plotted together for comparison. For both cases, a significant influence between ROIs was observed across the range of frequencies studied up to 50 Hz. To examine specific frequency band influences, the spectrum was segmented into four bands: theta (4–8 Hz), alpha (9–12 Hz), beta (13–30 Hz) and gamma (31–50 Hz). We have examined higher frequency bands, but no significant CGC effects were identified above 50 Hz. Average CGC magnitudes were computed for both directions between ROI pairs. Permutation tests were performed for each frequency band by constructing a null distribution of signed pre- and post-disconnection spectrally segmented differences following recalculation of spectral CGC after random assignment of time series to the two categories. To compare whether directional influences between ROIs change between pre-and post-disconnection, random permutation tests were performed on the absolute difference between spectral CGC across frequencies. Each post-stimulus trial time series was randomly assigned to either the pre- or post-disconnection category and spectral CGC recalculated 2000 times. An empirical null distribution was calculated by summing the absolute CGC differences between pre-and post-disconnection across frequencies. The number of permuted surrogates that exceeded the measured difference was used to calculate $p$-values. Multiple comparison correction was applied: false discovery rate (FDR) was controlled at the 0.05 level[124] across ROI pairs and frequency bands. Only FDR corrected directional edges showing differences between pre- and post-disconnection are shown in Fig. 4B.

## Graph theoretic measures

Node hubness can be regarded as a continuous variable that may be quantified for a given node by using graph theoretic metrics such as weighted degree centrality, defined as the total strength of links incident upon a specific node[51]. Here, we derived a measure of hubness from the weighted degree centrality of the functional connectivity data. Specifically, hubness was defined as the sum of each node's bidirectional (in and out) functional connectivity edge weights. These weights were taken directly from the results of the state-space CGC effective connectivity analysis. We tested the specific a priori hypothesis about the hub strength (HS) of TP relative to the grouped other ROIs [HS(HG), HS(STG) HS(IFG opercularis), HS(IFG triangularis)]. We used the median to summarize the central tendency across the ROIs as it is a robust measure to extreme surrogates generated by the edge-swapping permutations. The hypotheses then are:

H0: HS(TP) < = median([HS(HG), HS(STG)HS(IFGop), HS(IFGtri)])

H1: HS(TP) >median([HS(HG), HS(STG)HS(IFGop), HS(IFGtri)])

Four independent tests then remain across the 4 bands (theta, alpha, beta, gamma), requiring multiple comparison correction. The test statistic is HS(TP) - median([HS(HG), HS(STG) HS(IFGop), HS(IFGtri)]). The empirical null distribution is derived using network edge-swapping surrogates for this test statistic. The observed (actual) test statistic is compared to an empirical null distribution, and the surrogates that exceed the observed test statistic are counted and divided by the number of repetitions to derive the $p$-value. We used 100,000 repetitions to generate 100,000 surrogates. $P$-values were corrected with the Benjamini-Hochberg procedure[124] to control the

false discovery rate at a level of 0.05. The corrected results for TP are $p = 0.036$, therefore, the TP is significantly hub-like in the gamma band. For detailed results, see Supplementary Table 6.

## Minimizing anesthetic and direct neurosurgical confounds

To optimize the quality of clinical ECoG recordings and minimize the effects of anesthesia, surgical anesthesia had been discontinued in both participants at least 20 min before the pre- and post-disconnection recording periods. Vital signs were similar and stable during both pre- and post-disconnection periods (e.g., heart rate and respiration: Supplementary Table 4). The participants were awake during the recordings, and we were able to obtain data in P2 on a tone-detection paradigm during the speech priming task (Results). Immediate post-anesthesia drowsiness is expected, but both participants were monitored throughout the task and recording periods to ensure that they were alert and similarly engaged throughout both pre- and post-disconnection. Both recordings (pre- and post-disconnection) were conducted after the craniotomy procedure and were similarly affected by it. The ATL disconnection step in the surgical procedure can be expected to alter brain functions either through direct surgical impact on neural tissue, or as a result of edema formation in tissue adjacent to the surgical site[125–127]. Edema is an implausible cause of the reported fronto-temporal effects because these areas were centimeters away from the surgical disconnection site, but effects owing to diaschisis are reported (Results). The responses to the speech sounds are consistent with prior reports conducted in the awake state, in both form and amplitude[128–130], and we observed little impact by the ATL disconnection procedure on general auditory speech feature processes, such as the auditory cortex VOT representations (Fig. 3A).

## Minimizing chronic epilepsy network impact

The current study conducted direct pre- and post-disconnection comparison tens of minutes prior to and after the ATL disconnection procedure itself, with the pre-disconnection results acting as an important reference that incorporates any prior effects on the system. As with other studies in epilepsy patients the system cannot be considered normal. In relation to previous literature comparing the semantic networks of patients with chronic anterior temporal lobe epilepsy to healthy individuals using neuroimaging, Shimotake and colleagues[64] show that the area of maximal semantic activation in patients and controls is in the same locus. However, it may be that the maximal location of activation is unchanged, but that there is already some compensation and more reliance on other nodes in the network, and on bilateral representations (see ref. 131 for an argument along these lines). This would likely reduce changes seen after ATL disconnection, as the epilepsy would have in essence already induced compensation or a partial disruption/disconnection of the functional network. Thus, here we may not be seeing all of the changes that might occur from disconnecting an ATL that had not experienced seizures. However, acute changes as reported here shortly before and after the ATL disconnection procedure likely reflect the acute loss of core processes that have not been disrupted by chronic epilepsy or its causes. Therefore, findings of immediate alterations in the neural network would have occurred even in the face of possible pre-operative chronic compensatory processes. Also, in these patients, the pre-surgical neurophysiological system shows fairly normal structural connectivity and function in both patients, including speech-related and mismatch responses in HG and speech-related responses in IFG (Fig. 3B). We also only make comparisons between pre-resection and post-resection effects that are significant and consistent in both patients, a within-subjects comparison, rather than comparisons to data in controls. Regarding the reported behavioral and neurophysiological results, the neurophysiological results are much more immediate pre- and post-surgical comparisons conducted tens of

minutes before and after only the ATL disconnection step, whereas the behavioral results were only possible to obtain months before and after the neurosurgical procedure, which in one of the patients also involved the further MTL resection step. The pre-surgical neurophysiological responses serve as a more recent baseline comparison, taking into account any pre-existing network-level functional alteration and assessing only the additional impact of the loss of the ATL.

## Reporting summary

Further information on research design is available in the Nature Portfolio Reporting Summary linked to this article.

## Data availability

The datasets generated in this study have been deposited in the Zenodo database under accession code https://doi.org/10.5281/zenodo.8110724. Source data is provided with this paper. Source data are provided with this paper.

## Code availability

No custom software was used during the current study. Analysis scripts are available from the corresponding authors upon request.

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

## Acknowledgements

Supported by Wellcome Trust (CIP: WT092606AIA; TDG: WT091681MA), National Institutes of Health USA (MAH: R01–DC04290; U01-NP103780) and European Research Council (CIP: ERC CoG, MECHIDENT). PNT is supported by a UKRI Future Leaders Fellowship [MR/T04294X/1]. We thank Dr. R. Mueller and H. Chen for assistance with data collection, F. Balezeau for help with the dMRI processing, and T. Abel, M. Banks, M. Long, K.V. Nourski, H. Oya, B. Park and M. Steinschneider for discussion.

## Author contributions

Z.K.: Data collection, Conceptualization, Methodology, Software, Formal analysis, Visualization, Writing – original draft, review & editing. R.L.J.: Conceptualization, Methodology, Software, Formal analysis, Investigation, Visualization, Writing – original draft, review & editing. P.N.T.: Software, Formal analysis, Investigation, Writing – original draft, review & editing. R.M.C.: Visualization, Investigation, Writing – review & editing. B.M.: Conceptualization, Methodology, Writing – review & editing. A.E.R.: Conceptualization, Methodology, Writing – review & editing. M.E.S.: Data collection, Software, Conceptualization, Methodology, Writing – review & editing. C.D.S.: Data collection, Formal analysis, Writing – review & editing. Y.K.: Software, Formal analysis, Investigation, Writing – review & editing. P.E.G.: Investigation, Writing – review & editing. J.I.B.: Data collection, Formal analysis, Writing – review & editing. C.K.K.: Investigation, Writing – review & editing. I.C.: Investigation, Writing – review & editing. J.D.G.: Investigation, Writing – review & editing. H.K.: Investigation, Writing – review & editing. T.E.C.: Investigation, Writing – original draft, review & editing. T.D.G.: Conceptualization, Writing – review & editing, Supervision, Funding acquisition. M.A.H.: Conceptualization, Writing – review & editing, Supervision, Project administration, Funding acquisition. C.I.P.: Conceptualization, Methodology, Visualization, Investigation, Funding acquisition, Supervision, Writing – original draft, review & editing.

## Competing interests

The authors declare no competing interests.

## Additional information

[1]Department of Neurosurgery, University of Iowa, Iowa City, IA, USA. [2]Biosciences Institute, Newcastle University Medical School, Newcastle upon Tyne, UK. [3]Neuroscience Institute, Carnegie Mellon University, Pittsburgh, PA, USA. [4]Departments of Neuroscience and Psychology, University of Wisconsin, Madison, WI, USA. [5]CNNP Lab, Interdisciplinary Computing and Complex BioSystems Group, School of Computing, Newcastle University, Newcastle upon Tyne, UK. [6]UCL Institute of Neurology, Queen Square, London, UK. [7]Department of Psychological and Brain Science, University of Iowa, Iowa City, IA, USA. [8]Psychology Department, Gonzaga University, Spokane, WA, USA. [9]Department of Radiology, University of Iowa, Iowa City, IA, USA. [10]Iowa Neuroscience Institute, University of Iowa, Iowa City, IA, USA. [11]Department of Communication Sciences and Disorders, University of Iowa, Iowa City, IA, USA. [12]Department of Clinical Neurosciences, Cambridge University, Cambridge, UK. [13]MRC Cognition and Brain Sciences Unit, Cambridge University, Cambridge, UK. [14]These authors contributed equally: Thomas E. Cope, Timothy D. Griffiths, Matthew A. Howard III, Christopher I. Petkov.
✉e-mail: kocsis.zsuzsanna86@gmail.com; chris.petkov@ncl.ac.uk

