## [Peer Review File · Nature Communications]

Immediate Neural Impact and Incomplete Compensation After Semantic Hub DisconnectionREVIEWER COMMENTS

Reviewer #1 (Remarks to the Author):

The overall topic of explaining how the anterior temporal lobe supports language is interesting and the disconnection dataset is rare and potentially very important. The authors sought to test whether left ATL disconnection would cause 1) no impact, 2) solely disruption, or 3) partial compensation in fronto-temporal regions. The patterns of language-related electrophysiological shifts across the brain following ATL disconnection are interpreted as providing evidence in favor of the partial compensation hypothesis, which was tested using a predictive coding framework.

The data analyses in this paper are mostly well done, although some key plots are not explained/integrated. While I am enthusiastic about much of the work, I am concerned that the reported findings do not strongly support the author's primary claims for hypothesis #3 (partial compensation/predictive coding). The data are presented rather comprehensively (which is great) but there are a range of findings here that I do not believe provide specific support for the partial compensation hypothesis. Therefore, I find the authors' core interpretation to be a bit problematic/oversimplified given the complex pattern of results in their data. In light of this issue, I think that the authors should temper some of their claims and perhaps acknowledge the complexity of their findings. Below, I provide specifics beyond these ideas and I also provide other suggestions for how the authors could improve the clarity of their report. I also explain a couple of situations where there are data analyses that seem entirely unintegrated with the text, which disrupts the flow of the paper.

Major comments:

* The claim on line 180 is problematic: "The results show that before ATL disconnection the temporal pole contacts in this region are hub-like within the recorded network in beta and gamma frequencies". I do not see any justification in figure 3D or supp figure 4 for the claim that the TP region is distinctive in terms of connectivity for beta and gamma oscillations. Instead it seems that connectivity is greater overall at low frequencies and that there is simply greater CGC strength overall for HG and TP at all frequencies. Also there is no statistical evidence for this claim, as the results seem to be largely qualitative based on the shape of the bar charts. Statistical justification is necessary for such claims. Also, finally, it is confusing that the text references Figure 3D because that figure does not mention either beta or gamma bands.

* The right panel of Figure 3A is extremely hard to read due to the number of overlapping lines and the similarity of colors chosen to represent different conditions. Thus, I cannot evaluate their claim that the

timing of the auditory cortical responses was preserved. I suggest that the authors find a clearer way to explain this key point.

* Figure 3C is cited to support the point that “the neurophysiological semantic mismatch response was depleted post-disconnection in HG”. However, in fact the data in Figure 3G does not really support this point because there are significant effects in both directions at varying timepoints. I have no idea how to interpret the complex pattern of results in this plot because at some timepoints the blue line is significantly above the red line and at other timepoints the reverse pattern is significantly present. At a minimum, the pattern of results seems substantially more complex than the authors claim.

* Figure 4 is rarely discussed in the paper. I am not sure how it fits into the story but in its current form the flow seems rather disrupted.

* Regarding the results shown in Figure 5: it is intriguing that there is increased connectivity in gamma post disconnection. However, can this result be distinguished from changes in power across frequency bands? In ECoG recordings, oscillatory power and connectivity can be confounded with each other. So to support the claim of changes in connectivity following disconnection it would be necessary to distinguish the results from changes in oscillatory power.

* Supplemental figure 3B is hard to interpret because this method is novel for ECoG data. It would be important to show that these results are not related to the locations of the electrodes that were implanted in each patient, as well as other potential confounds.

* The claims about supp figure 6 that the IFG components were disrupted post disconnection are hard to understand (lines 237) because the lines in the plots are rather variable, sometimes showing the red line above the blue and sometimes the reverse. Is there a correction for multiple comparisons here? I find this plot entirely impossible to interpret. I also have a similar concern regarding the results in Supp figure 7 and the associated text.

Minor points:

* The link between predictive coding theory and distinct oscillation bands need to be more clearly stated and justified in terms of the underlying physiology.

* The hubness measure appears in multiple sections of the paper (figures 3D, 4 and 5). A clear visualization of the hubness on the connectivity graph (like node-size) would be beneficial in all corresponding panels.

* A methodological concern: Only electrodes with significant speech responses were evaluated. Was this selection procedure applied independently for pre and post implantation recordings?

Reviewer #2 (Remarks to the Author):

The manuscript explores the important issue of how connectivity and language comprehension change as a result of lesions to the anterior temporal lobe, an area hypothesized to be central to semantic processing. Two patients underwent invasive recordings during a task in which the semantic content in the early part of the sentence biased the meaning in the later part of the sentence, allowing for detection of semantic match or mismatch. The authors find that there are disruptions in the task following lesions to the ATL and changes in network connectivity/gamma band responses as a result of the disconnection. The results showed fairly clearly that connectivity patterns may have increased between areas with prefrontal cortex with other areas showing decreases. There were also findings of changes for congruent vs. incongruent speech processing as a result of disconnection in terms of (potential) differences in ERPs. The paper concludes with support for the predictive coding model.

Assessment: This manuscript capitalizes on an extremely rare opportunity to look at changes before and after a brain lesion using electrophysiology shortly before and shortly after the lesion. The task is well-designed and appropriate for the question at hand, although more behavioral work up could be helpful. The major concerns here centered around 1) the diminished focus on low-frequency oscillations, which are now better accepted as an inter-regional communicator than gamma 2) the use of granger causality 3) lack of statistics to clearly support pre vs. post differences, in some instances 4) somewhat unclear traction on the diaschisis vs. predictive coding model. Overall, however, I am enthusiastic about the contributions of this manuscript and with some additional analyses and consideration, believe this manuscript can be suitable for publication at NC.

MAJOR

1. Right now, the presentation focuses on a fairly large set of analyses across many different frequency bands and using ERPs. While I think it is important to perform these analyses in some form and report them somewhere, it makes the results a little hard to follow and gives a feeling of lack of focus. The analyses, to some extent, focus on gamma as an inter-regional coupling signal but perhaps this can

refocused slightly. There is now increasing consensus in the memory / ECoG field, that lower frequencies signals are instead likely to serve this function (Solomon et al., 2017; Watrous, Tandon, Conner, Pieters, & Ekstrom, 2013). I think this issue is also present in the in this manuscript (Figure 4); gamma shows generally less coupling than lower frequencies, with high gamma showing essentially no coupling. I think there are also good theoretical arguments to point out that gamma is simply too fast to play a meaningful role in inter-regional coherence and instead reflect multiunit firing patterns locally (Ray & Maunsell, 2010; Shadlen & Movshon, 1999). My suggestion then would be to refocus some of the analyses regarding inter-regional coupling on lower-frequency signals. It might also be beneficial to consider some of the memory literature mentioned above (and numerous other papers showing something similar with regard to memory processing and low-frequency oscillations) as means of refocusing the networking analyses.

2. The granger causality analysis is arguably not the optimal approach. One issue with granger methods is that they can be sensitive to lags introduced through third sources. It is not clear that the authors can rule out a third source in their analyses and more generally, the directionality in graph theory analyses is not typically paramount. I might suggest trying phase synchronization methods instead to derive graph theory measures (Vinck, van Wingerden, Womelsdorf, Fries, & Pennartz, 2010). At least, in the current form, the granger methods are not particularly well justified and could involve problematic assumptions that are not testable (i.e., third sources and other sources involving volume conduction producing unknown lags).

3. The statistics generally focus on either pre or post but few of the stats focus on changes from pre to post. For example, Figure 3b and c would benefit from marked areas showing the difference in the signals. Also, for Figure 3c, it is claimed that the signal is “depleted” but no comparison is performed against zero.

4. A major thrust of the paper is testing theoretical models of network rearrangement. The paper tests the ideas of no impact, diaschisis, and predictive coding. While the paper certainly rules out the no impact hypotheses (as would other work that is already out there), it is not clear that the data can really adjudicate between diaschisis and predictive coding. There are clearly changes in gamma in IFG, wouldn't this be an example of a distal connected site that is affected? Also, more citations and scholarly presentation of diaschisis is probably needed, perhaps these papers can be a useful start: (Carrera & Tononi, 2014; Finger, Koehler, & Jagella, 2004; Henson et al., 2016). Perhaps this issue simply needs more scholarly consideration. I also think it would be useful to explain the hub and spokes model in a little more detail. Finally, along these lines, while the behavioral data from the task was fairly clear, I wondered about other neuropsych tests of semantic naming, lexical decision, and verbal fluency. Were these looked at via neuropsychs? Did these change in anyway pre to post lesion?

MINOR

1. Figure 4A was difficult to follow. What is influencing what?

2. It was not clear what the columns are in Figure 4B. Could the gamma-related “connectivity” changes be the result of aberrant discharge?

3. The predictions in Figure 1 should also include predictions for lower-frequency coherence, given what is discussed in Major point 1.

4. "Theta oscillations currently have a more ambiguous role in the predictive coding framework, having been associated with both feedforward entrainment to the speech syllabic rate, cognitive sampling at a theta rhythm and feedback processes " -> I am not sure if theta oscillations would be considered to have an ambiguous role in inter-regional interactions. I suggest the authors consider some of the work cited in Major point 1 and consider these ideas in a little more detail.

5. "The findings indicate that the left ATL is likely to act in two key ways: 1) as a higher level site in a constructive hierarchy" -> I am not sure how the results demonstrate a hierarchy here. IFG and other areas may play partially compensatory roles. How does this show a hierarchy?

6. "Thereby, it is important for future studies to establish direct correspondence to source localized EEG or fMRI data in the same patients." -> Scalp EEG, with a localization error using source reconstruction around 1cm^3 , cannot definitely identify deep brain signals. It is not clear how it could be used for those purposes here.

References

- Carrera, E., & Tononi, G. (2014). Diaschisis: past, present, future. *Brain*, 137(Pt 9), 2408-2422. doi:10.1093/brain/awu101
- Finger, S., Koehler, P. J., & Jagella, C. (2004). The Monakow concept of diaschisis: origins and perspectives. *Arch Neurol*, 61(2), 283-288. doi:10.1001/archneur.61.2.283
- Henson, R. N., Greve, A., Cooper, E., Gregori, M., Simons, J. S., Geerligs, L., . . . Browne, G. (2016). The effects of hippocampal lesions on MRI measures of structural and functional connectivity. *Hippocampus*, 26(11), 1447-1463. doi:10.1002/hipo.22621
- Ray, S., & Maunsell, J. H. (2010). Differences in gamma frequencies across visual cortex restrict their possible use in computation. *Neuron*, 67(5), 885-896. doi:S0896-6273(10)00614-8 [pii] 10.1016/j.neuron.2010.08.004
- Shadlen, M. N., & Movshon, J. A. (1999). Synchrony unbound: a critical evaluation of the temporal binding hypothesis. *Neuron*, 24(1), 67-77, 111-125. doi:S0896-6273(00)80822-3 [pii]
- Solomon, E. A., Kragel, J. E., Sperling, M. R., Sharan, A., Worrell, G., Kucewicz, M., . . . Kahana, M. J. (2017). Widespread theta synchrony and high-frequency desynchronization underlies enhanced cognition. *Nat Commun*, 8(1), 1704. doi:10.1038/s41467-017-01763-2
- Vinck, M., van Wingerden, M., Womelsdorf, T., Fries, P., & Pennartz, C. M. (2010). The pairwise phase consistency: a bias-free measure of rhythmic neuronal synchronization. *Neuroimage*, 51(1), 112-122. doi:10.1016/j.neuroimage.2010.01.073

Watrous, A. J., Tandon, N., Conner, C. R., Pieters, T., & Ekstrom, A. D. (2013). Frequency-specific network connectivity increases underlie accurate spatiotemporal memory retrieval. *Nature neuroscience*, 16(3), 349-356. doi:10.1038/nn.3315

Reviewer #3 (Remarks to the Author):

The authors of the present manuscript tested the necessity of anterior temporal lobe (ATL) for semantic processing and its role as a "hub" in semantic processing. Two patients with intracranially implanted electrodes participated and EEG data were collected pre- and post- ATL-disconnection surgery with a specific focus on inferior frontal gyrus (IFG), Heschel's Gyrus (HG) and superior temporal sulcus (STS). Patients performed active and passive versions of a semantic prediction task in which participants listen to sentences that prime perception of a final target word. The authors find changes in spectral signals as well as connectivity in the comparison of pre- to post-surgical data. The authors state that their findings are consistent with the predictive coding framework, wherein ATL is responsible for providing predictions via feedback connections to downstream perceptual regions (e.g. HG) and that in the absence of ATL, other regions such as IFG engage with sensory regions differently to compensate.

The question motivating the study -- necessity and role of ATL in semantic processing -- would be of interest to the field. However, the many methodological issues and conflicting results limit my enthusiasm for the manuscript and do not clearly support the conclusions. I detail these concerns below.

1. Inconsistent findings that are challenging to interpret

- Pre-surgery patient performance is not comparable to control performance, suggesting that semantic processing is already disrupted in these patients even before ATL disconnection. Thus, it is hard to conclude that the changes observed in the present study reflect how mechanisms might change following ATL loss in other contexts. Indeed, given the difference between patients and controls, patients may already be engaging compensatory mechanisms on account of their epilepsy, and thus the IFG changes observed are not a response to ATL loss specifically, but the combination of other resection (see below) and ATL disconnection. IFG changes may not be an "immediate" response if other deficits had arisen prior to ATL disconnection. The authors do note that this is a within-subjects case of cases study; however, the ability of the present findings to inform other work seems even more limited given this behavioral discrepancy.

- The main effect of disconnection on the behavioral responses occurs in opposite directions across the two patients. The authors suggest that this may be due to additional medial temporal lobe resection in patient 1. At the very least, it seems that this cannot be described as a main effect of ATL disconnection. Why disconnection might improve performance is also challenging to understand. Finally, these findings, and the fact that one patient had a more extensive resection, suggest that pre-surgery performance and neural signals are unlikely to reflect typical neural processes or mechanisms.

- The authors state, "one of the few neurophysiological signatures that remained largely unaffected pre- and post-disconnection in both participants was the preserved timing of the auditory cortical responses to the VOTs." This claim is also repeated in the discussion. However, the authors report a significant effect for Peak 2 + patient 2. It is challenging to compare pre- vs. post- disconnection in Figure 3 as the ERPs are presented in separate panels.

2. Multiple methodological details that confound the results or lack clarity.

- A number of experimental details are missing and as such it is challenging to understand how patients form expectations in the semantic prediction task. It appears that the same sentence is repeated 24 times in a session, as suggested by the text stating that 12 sentences were used over 288 trials. I would infer from "The target word was manipulated along a VOT continuum with 6 steps" that there were four trials each with each stepped VOT, but this was not made explicit. If that is the case, why would participants predict the semantically congruent word? Especially considering that there were multiple sessions with what appears to be the same stimuli (since the pre-surgery sessions were used to select stimuli for the surgery sessions), participants should learn to expect other words than the semantically congruent one. In the absence of predictions, and possibly when predictions are modified by experimental parameters, the role of a putative hub region such as ATL will necessarily be different than it would be for more naturalistic semantic predictions.

- Putting aside the above issue, if patients are predicting the semantically congruent word prior to target presentation, using the time prior to target presentation as a baseline is inappropriate. Patients should be making predictions during this time interval, it is not "neutral" (i.e. it is not a true baseline interval, if one exists) and will artificially create effects during the target interval. This impacts both the selection of the "speech responsive" electrodes and pre/post surgery comparisons. An appropriate pre-stimulus baseline would be a time interval before a trial (sentence) begins, when patients have no expectations/preparation. Alternatively, the data could be z-scored across all trials, which would obviate the need for baseline interval selection (which appears to be what the authors did for the state-space conditional granger causality analysis).

- There is the potential that changes in pre- vs. post-surgery signals are driven by the selection of different electrodes in each condition.

- No justification is provided for frequency band selection and bands do not match across analyses (Figure 3 vs. Figure 5). The only written definition of the bands appears at the end of the methods. In general, the authors make broad statements about frequency that are not clearly motivated (e.g. "beta/alpha < 30 Hz" on page 5; multiple frequencies fall below 30 Hz). It is unclear why the authors expected strong broadband activity to speech sounds (Page 8), especially considering evidence for high frequency increases and low frequency decreases during auditory perception (Crone et al. (2001) Induced electrocorticographic gamma activity during auditory perception. *Clinical Neurophysiology*).

- More information is needed as to how centrality, hubness, and Conditional Granger Causality strength are determined. There should be explicit values provided, based on prior work, that must be observed for a region to be defined as "hub-like." How regions of interest (ROIs) were defined (i.e. how electrode inclusion in an ROI was determined) and the number of electrodes per ROI is not reported. It is unclear why both patients' data were concatenated seeing as patients performed differently on the behavioral measures.

3. Logic and motivation of the hypothesis; interpretation of the findings.

The authors propose that IFG may fill a compensatory role, "Other regions could compensate for an impact on the ATL, or can maintain established semantic representations, at least temporarily. Candidate compensatory regions would include the language-related hub in the left inferior frontal gyrus (IFG)" (Page 2). However, if IFG is performing a compensatory role, why would feedback signals (alpha/beta) be disrupted after ATL disconnection (Figure 1)? Shouldn't feedback signals increase if IFG is compensating for the loss of ATL? Broadly speaking, the hypotheses feel a bit post-hoc and although the prediction framework makes sense on its own, the specific predictions made (i.e. disruptions to feedback signals) do not seem to match this hypothesis. Likewise the findings do not clearly support the predictive coding framework, given the issues with prediction outlined above.

Reviewer Point-by-point Replies

We have found every comment useful and in this major revision of the submitted manuscript we have worked to strengthen the manuscript by addressing the reviewer comments as suggested. This is given in detail further below in the point-by-point replies to each reviewer comment.

Reviewer #1 Comments

“The overall topic of explaining how the anterior temporal lobe supports language is interesting and the disconnection dataset is rare and potentially very important. The authors sought to test whether left ATL disconnection would cause 1) no impact, 2) solely disruption, or 3) partial compensation in fronto-temporal regions. The patterns of language-related electrophysiological shifts across the brain following ATL disconnection are interpreted as providing evidence in favor of the partial compensation hypothesis, which was tested using a predictive coding framework. The data analyses in this paper are mostly well done, although some key plots are not explained/integrated. While I am enthusiastic about much of the work, I am concerned that the reported findings do not strongly support the author’s primary claims for hypothesis #3 (partial compensation/predictive coding). The data are presented rather comprehensively (which is great) but there are a range of findings here that I do not believe provide specific support for the partial compensation hypothesis. Therefore, I find the authors’ core interpretation to be a bit problematic/oversimplified given the complex pattern of results in their data. In light of this issue, I think that the authors should temper some of their claims and perhaps acknowledge the complexity of their findings. Below, I provide specifics beyond these ideas and I also provide other suggestions for how the authors could improve the clarity of their report. I also explain a couple of situations where there are data analyses that seem entirely unintegrated with the text, which disrupts the flow of the paper.”

Reply: We appreciate the insightful comments and guidance on clarifying and strengthening the manuscript. We have worked to implement all of the suggested changes, detailed as follows.

Major comments:

Comment 1: *“The claim on line 180 is problematic: “The results show that before ATL disconnection the temporal pole contacts in this region are hub-like within the recorded network in beta and gamma frequencies”. I do not see any justification in figure 3D or supp figure 4 for the claim that the TP region is distinctive in terms of connectivity for beta and gamma oscillations. Instead it seems that connectivity is greater overall at low frequencies and that there is simply greater CGC strength overall for HG and TP at all frequencies. Also there is no statistical evidence for this claim, as the results seem to be largely qualitative based on the shape of the bar charts. Statistical justification is necessary for such claims. Also, finally, it is confusing that the text references Figure 3D because that figure does not mention either beta or gamma bands.”*

Reply: We have clarified this statement as suggested and statistically tested the claim about the hubness of the temporal pole (TP), as follows: Significance testing was calculated by randomly swapping edges of the directed weighted networks generating 10,000 surrogate networks to serve as an empirical null distribution for the sum of in- and outflow, which can then be used to compute p-values for the five nodes. As now reported in the manuscript (lines 202-210 and Fig. 3D) with this test only the TP contacts reached significance in the gamma band ($p = 0.018$).

Comment 2: *“The right panel of Figure 3A is extremely hard to read due to the number of overlapping lines and the similarity of colors chosen to represent different conditions. Thus, I cannot evaluate their*

claim that the timing of the auditory cortical responses was preserved. I suggest that the authors find a clearer way to explain this key point.”

Reply: We have improved the presentation of Figure 3A as follows:

(A) Speech VOT responses largely intact post-disconnection

To the right in blue lines and error bars are shown the pre-disconnection VOT Peak 1 and Peak 2 mean responses with variability of the data averaged across patients (4 electrodes averaged for each). Red lines show the post-disconnection responses, which are highly overlapping. The data to generate these plots are provided (Supp. Table 3). We have also ensured that the statements on the statistical testing in each of the subjects are accurate on the VOT responses (from line 245).

Comment 3: “Figure 3C is cited to support the point that “the neurophysiological semantic mismatch response was depleted post-disconnection in HG”. However, in fact the data in Figure 3G does not really support this point because there are significant effects in both directions at varying timepoints. I have no idea how to interpret the complex pattern of results in this plot because at some timepoints the blue line is significantly above the red line and at other timepoints the reverse pattern is significantly present. At a minimum, the pattern of results seems substantially more complex than the authors claim.”

Reply: We apologize for the lack of clarity in the presentation of these results. We have worked to both clarify their presentation in the text and to ensure the interpretation is accurate: The pre-disconnection (blue) waveform has significant deviation from zero in both positive and negative directions. This is abolished in the post-disconnection (red) trace, which does not significantly deviate from zero. Starting at line 266, “The pre-disconnection congruency mismatch response is seen in Fig. 3C as a difference wave (congruent versus incongruent; blue line) shows the expected larger congruent/incongruent responses throughout the target word, similar to a prior study (Cope et al., 2017). The post-disconnection response is significantly reduced in both patients (gray bars in Fig. 3C), seen as post-disconnection mismatch responses close to 0 difference (red line in Fig. 3C).” The figure panel is pasted next for convenience.

(C) Congruent vs. incongruent speech prediction effects

Additionally, we have tested both the pre- and post-disconnection response curves against zero (with cluster-based permutation testing, $p < 0.05$), and only the pre-disconnection mismatch response shows a response that differs significantly from zero for both the early and the later congruency (greater than incongruency) responses. These periods are also significantly different pre- and post-disconnection as reported in (Fig. 3C), supporting the conclusion in the text of a significantly reduced congruency mismatch response post-disconnection (lines 264-271).

Comment 4: “Figure 4 is rarely discussed in the paper. I am not sure how it fits into the story but in its current form the flow seems rather disrupted.”

Reply: We agree with the reviewer. This figure was originally provided to present the basis analysis approach for the prior Fig. 5 (currently Fig. 4), but since it breaks the flow of the text and does not provide crucial information, we have moved it to Suppl. Fig. 7.

Comment 5: “Regarding the results shown in Figure 5: it is intriguing that there is increased connectivity in gamma post disconnection. However, can this result be distinguished from changes in power across frequency bands? In ECoG recordings, oscillatory power and connectivity can be confounded with each other. So to support the claim of changes in connectivity following disconnection it would be necessary to distinguish the results from changes in oscillatory power.”

Reply: This is an insightful point and in Supplementary Fig. 4 we now also show the power changes post-disconnection. Suppl. Fig. 4 shows no evidence for significantly enhanced gamma power responses post-disconnection (all red traces are below the blue). This is the case for all of the recorded regions, excepting the magnified high gamma signal to the target speech sound in Heschl’s gyrus, one of the key reported results in the paper (Fig. 3B). By contrast, the increased connectivity changes in Fig. 4 occur mostly within the subregions of the IFG and STG, where gamma magnitude responses are not significantly stronger post-disconnection. We have noted these results in the text (line 282-287).

Comment 6: “Supplemental figure 3B is hard to interpret because this method is novel for ECoG data. It would be important to show that these results are not related to the locations of the electrodes that were implanted in each patient, as well as other potential confounds.”

Reply: We apologize for the confusion. The diffusion-weighted MRI connectivity results shown in Supplementary Figure 3B are not a new method. We have clarified the text and figure legend to ensure readers are clear on these being diffusion MRI (dMRI) data that were obtained 2-6 weeks before the procedure and compared to dMRI data obtained within 2 months after the surgical procedure.

Comment 7: *“The claims about supp figure 6 that the IFG components were disrupted post disconnection are hard to understand (lines 237) because the lines in the plots are rather variable, sometimes showing the red line above the blue and sometimes the reverse. Is there a correction for multiple comparisons here? I find this plot entirely impossible to interpret. I also have a similar concern regarding the results in Supp figure 7 and the associated text.”*

Reply: Suppl. Fig. 4 (previously Suppl. Fig. 5), we agree, was not clearly described in the text and we have ensured that descriptions throughout the manuscript are based on significant effects seen in both patients. We do not interpret effects that are too variable, not significant, or inconsistent across the two participants. See Results text lines 257-261 for the revised Suppl. Fig. 4 description: *“Higher-level stages in the auditory cortical hierarchy (STG) had largely reduced post-disconnection responses to the target word, in both participants, particularly for the gamma band (compare blue (pre) versus red (post) traces in Suppl. Fig. 4 for the STG).”* We have also revised to ensure that Suppl. Fig. 5 (complementing Fig. 3C) is only interpreted for the HG results, which are consistent and significant for both participants (see Suppl. Fig. 5 and manuscript Fig. 3C). Regarding the question about multiple corrections, cluster-based permutation is corrected for multiple comparisons at the cluster level (e.g., Maris and Oostenveld, 2007), we only consider significant differences longer than 25 ms in duration, and we had strong *a priori* hypotheses (Fig. 1) for the auditory cortical results tested and shown in Fig. 3.

Minor points:

Comment 8: *“The link between predictive coding theory and distinct oscillation bands need to be more clearly stated and justified in terms of the underlying physiology.”*

Reply: We have revised the text in the Introduction as suggested to make the link between the theory and distinct oscillation bands clearer in terms of the typically associated oscillatory frequencies and the underlying physiology (starting at line 101).

Comment 9: *“The hubness measure appears in multiple sections of the paper (figures 3D, 4 and 5). A clear visualization of the hubness on the connectivity graph (like node-size) would be beneficial in all corresponding panels.”*

Reply: We have revised Fig. 3D, Fig. 4 (prior Fig. 5), Suppl. Fig. 7 (prior Fig. 4) and Suppl. Fig. 6 so that they all now show node hub size by the size of the circle at each node.

Comment 10: *“A methodological concern: Only electrodes with significant speech responses were evaluated. Was this selection procedure applied independently for pre and post implantation recordings?”*

Reply: The selection procedure was only conducted for pre-disconnection recordings to identify the contacts to include for analysis, because disruption in the post-disconnection responses would not make them suitable to select the contacts. We have ensured that the methods state this clearly (starting line 717): *“Moreover, all of the speech responsive contacts identified pre-disconnection remained significantly speech responsive post-disconnection, with only one additional recording contact in P2 in the STG showing a significant speech responsive after disconnection that was not significant pre-disconnection.”*

Additional references cited here:

Cope, T. E. *et al.* Evidence for causal top-down frontal contributions to predictive processes in speech perception. *Nat. Commun.* **8**, 2154 (2017).

Maris, E. & Oostenveld, R. Nonparametric statistical testing of EEG- and MEG-data. *J. Neurosci. Methods* **164**, 177–190 (2007).

Reviewer #2

“The manuscript explores the important issue of how connectivity and language comprehension change as a result of lesions to the anterior temporal lobe, an area hypothesized to be central to semantic processing. Two patients underwent invasive recordings during a task in which the semantic content in the early part of the sentence biased the meaning in the later part of the sentence, allowing for detection of semantic match or mismatch. The authors find that there are disruptions in the task following lesions to the ATL and changes in network connectivity/gamma band responses as a result of the disconnection. The results showed fairly clearly that connectivity patterns may have increased between areas with prefrontal cortex with other areas showing decreases. There were also findings of changes for congruent vs. incongruent speech processing as a result of disconnection in terms of (potential) differences in ERPs. The paper concludes with support for the predictive coding model.

Assessment: This manuscript capitalizes on an extremely rare opportunity to look at changes before and after a brain lesion using electrophysiology shortly before and shortly after the lesion. The task is well-designed and appropriate for the question at hand, although more behavioral work up could be helpful. The major concerns here centered around 1) the diminished focus on low-frequency oscillations, which are now better accepted as an inter-regional communicator than gamma 2) the use of granger causality 3) lack of statistics to clearly support pre vs. post differences, in some instances 4) somewhat unclear traction on the diaschisis vs. predictive coding model. Overall, however, I am enthusiastic about the contributions of this manuscript and with some additional analyses and consideration, believe this manuscript can be suitable for publication at NC.”

Reply. We thank the reviewer for their assessment and comments. As detailed below, we have 1) ensured a stronger focus on the low-frequency oscillations throughout the manuscript as better inter-regional communication than gamma, 2) strengthened the justification of the state-space connectivity analysis approach and its robustness to indirect sources and volume conduction effects, 3) ensured the noted pre and post differences are supported statistically (see the point-by-point replies below), including providing the requested neuropsychological testing results, and, 4) incorporated the modern diaschisis hypothesis (our prior version only considered classical diaschisis) including integrating the recommended papers, which were relevant and useful to cite.

Major comments:

Comment 1: *“Right now, the presentation focuses on a fairly large set of analyses across many different frequency bands and using ERPs. While I think it is important to perform these analyses in some form and report them somewhere, it makes the results a little hard to follow and gives a feeling of lack of focus. The analyses, to some extent, focus on gamma as an inter-regional coupling signal but perhaps this can refocused slightly. There is now increasing consensus in the memory / ECoG field, that lower frequencies signals are instead likely to serve this function (Solomon et al., 2017; Watrous, Tandon, Conner, Pieters, & Ekstrom, 2013). I think this issue is also present in the in this manuscript (Figure 4); gamma shows generally less coupling than lower frequencies, with high gamma showing essentially no coupling. I think there are also good theoretical arguments to point out that gamma is simply too fast to play a meaningful role in inter-regional coherence and instead reflect multiunit firing patterns locally (Ray & Maunsell, 2010; Shadlen & Movshon, 1999). My suggestion then would be to refocus some of the analyses regarding inter-regional coupling on lower-frequency signals. It might also be beneficial to consider some of the memory literature mentioned above (and numerous other papers showing something similar with regard to memory processing and low-frequency oscillations) as means of refocusing the networking analyses. “*

Reply. We have streamlined the presentation in the main manuscript (moving Fig. 4 to the supplement, now Suppl. Fig. 7) and have cut two supplementary figures (Suppl. Fig. 6 and 8.

previously) that were not thoroughly interpreted and were not necessary for interpretation. We did not intend to overly focus on gamma, and completely agree the impact on theta interconnectivity in the results is if anything more striking than for gamma (Fig. 4; lines 252-263). We have revised the manuscript to ensure a more balanced presentation of the lower frequencies, in particular theta oscillations (lines 344-348) and have cited and considered the link to the broader memory literature on theta-based interconnectivity for cognition. Furthermore, we have ensured that hypothesis Fig. 1 has the appropriate *a priori* theta band predictions integrated.

Comment 2: *“The granger causality analysis is arguably not the optimal approach. One issue with granger methods is that they can be sensitive to lags introduced through third sources. It is not clear that the authors can rule out a third source in their analyses and more generally, the directionality in graph theory analyses is not typically paramount. I might suggest trying phase synchronization methods instead to derive graph theory measures (Vinck, van Wingerden, Womelsdorf, Fries, & Pennartz, 2010). At least, in the current form, the granger methods are not particularly well justified and could involve problematic assumptions that are not testable (i.e., third sources and other sources involving volume conduction producing unknown lags).”*

Reply. We appreciate the opportunity to better justify using the state-space CGC approach and to address concerns related to third sources and volume conduction. The following text has been added to the Methods (starting at line 779). *“The state-space variant of Granger Causality was used because of the necessity for assessing directional effects conditional on the other recorded nodes and reference electrodes and its capability to deal with indirect volume conduction artifacts. The approach is conditional in the sense that simultaneous time series from a collection of electrodes are included to account for direct and indirect influences between electrodes. Although accounting for all possible indirect sources is not possible if those are not directly recorded, the intent is to account for a subset of indirect sources and effects upon them by conditioning. There is a concern that indirect sources and volume conduction could induce instantaneous correlated noise across recording electrodes with a lag that appears like effective connectivity. To minimize this concern, we routinely quality control the data by, for example, re-referencing to ground or other contacts. Also, the state-space method of calculating CGC has been shown to be more robust to the effects of volume conduction common to EEG analysis. For instance, simulation studies (Van de Steen et al., 2019) show that state-space CGC (Barnett & Seth, 2015) is able to model volume conduction and the evolution of the neural state within the same framework. These studies show that the state-space approach can result in the correct recovery of the original causal structure of the dynamical system in the presence of volume conducting noise. Cheung et al. (2010) also show that CGC analysis using state-space models was less sensitive to observation noise compared with two-stage VAR-based models. Faes and colleagues (2017) show that the state-space method yielded highly accurate spectral estimates of CGC that followed expected profiles over coupled directions and negligible magnitudes over uncoupled nodes. The state-space CGC estimator also showed higher reliability compared with standard VAR-based methods.”*

Comment 3: *“The statistics generally focus on either pre or post but few of the stats focus on changes from pre to post. For example, Figure 3b and c would benefit from marked areas showing the difference in the signals. Also, for Figure 3c, it is claimed that the signal is “depleted” but no comparison is performed against zero.”*

Reply: We have ensured that these pre/post statistical comparisons are available, as suggested. Namely, in Figure 3B (line plots to the right) the green bars show where there is significant difference between the pre- and post-disconnection conditions (cluster-based permutation test, $p < 0.001$). In Fig. 3C the gray bars depict the significant difference at the given time points between the mismatch response pre- and post-disconnection. We have also checked the mismatch responses against zero and only the pre-disconnection mismatch response is significant (cluster-based

permutation test relative to 0, $p < 0.05$). See the reply to Reviewer 1 Comment 3. Furthermore, we have replaced the word “depleted” with significantly “diminished” (line 265 and 325).

Comment 4: *“A major thrust of the paper is testing theoretical models of network rearrangement. The paper tests the ideas of no impact, diaschisis, and predictive coding. While the paper certainly rules out the no impact hypotheses (as would other work that is already out there), it is not clear that the data can really adjudicate between diaschisis and predictive coding. There are clearly changes in gamma in IFG, wouldn't this be an example of a distal connected site that is affected? Also, more citations and scholarly presentation of diaschisis is probably needed, perhaps these papers can be a useful start: (Carrera & Tononi, 2014; Finger, Koehler, & Jagella, 2004; Henson et al., 2016). Perhaps this issue simply needs more scholarly consideration. I also think it would be useful to explain the hub and spokes model in a little more detail. “*

Reply: We appreciate the opportunity to better consider the diaschisis and predictive coding accounts. Our prior version only considered the classical diaschisis hypothesis of functional disruption of interconnected sites. The references cited by the reviewer have been useful to integrate the modern diaschisis hypothesis alongside the predictive coding framework (see revised Introduction starting at line 101). The modern diaschisis framework, as the reviewer notes, accommodates functional increases in activity, increased interconnectivity, and changes in network properties, as we have seen in this study. That said, the predictive coding framework was more specific with regards to the form of the alterations in several ways as noted in the revised Discussion (starting at line 321). Nonetheless, these accounts are not contradictory, and we acknowledge that the modern diaschisis framework is useful from a more general perspective. The results with this acute causal effect on the semantic network support and extend both frameworks, and we find that the paper is now much more broadly interesting considering the modern diaschisis perspective.

Comment 5: *“Finally, along these lines, while the behavioral data from the task was fairly clear, I wondered about other neuropsych tests of semantic naming, lexical decision, and verbal fluency. Were these looked at via neuropsychs? Did these change in anyway pre to post lesion?”*

Reply: As suggested, we have added additional information in Suppl. Table 2 on the neuropsychological tests that were conducted in both patients. Reliable Change Index (RCI) results and population norm (z-scores, see Methods, lines 601-621) are presented in Suppl. Table 2 for all tests that the patients completed before and after the surgery day. We have added this information to the Results (lines 180-191) on the results with the neuropsychological tests: *“Post-operatively, both participants showed significant improvement in short-term memory for faces (WMS-III Faces I; RCI: P1: 3.402, P2: 2.91), which is not unexpected in MTL epilepsy patients (Baxendale et al., 2016). No other tests showed significant changes consistently across both participants at the time of testing. Two language-related tests were administered to both participants, however, only pre-operatively: the Boston Naming Test (BNT) which is a test for naming drawings of objects, and the Controlled Oral Word Association (COWA) Test, which is a verbal fluency test. On the latter, none of the participants were impaired pre-operatively, and on the BNT, Participant 1 was right on the cusp of impairment (score of 48), while Participant 2 was in the average range (score of 55).”*

MINOR

Comment 6: *“Figure 4A was difficult to follow. What is influencing what?”*

Reply: We have moved this figure to Suppl. Figure 7 because it was a Methods figure and broke the manuscript flow. It was meant to show the data that the summary state-space CGC results in Figure 5 (now Figure 4) are based on. This figure shows the CGC connectivity matrix across the

spectral frequencies with the significance testing results (A) and the hubness measure results across the nodes in (B). It does work better as a supplementary figure.

Comment 7: *“It was not clear what the columns are in Figure 4B. Could the gamma-related “connectivity” changes be the result of aberrant discharge?”*

Reply: The columns in Figure 4B (now Suppl. Fig. 7) depict the directional influences between regions of interest in each frequency band. The thickness of the lines represents the strength of the connection. As for the “aberrant discharge”, we do not think that the CGC results can be easily explained by power increase such as an aberrant gamma discharge, in part because gamma power effects were only significant in HG, and because, by contrast, the effective connectivity impact is stronger in the lower frequencies (e.g., theta). Also, see reply to the reviewer’s Comment 1 and the reply to Reviewer 1 Comment 5, which asks whether the connectivity results are related to oscillatory power: Suppl. Fig. 4 results when compared to Fig. 4 on interconnectivity suggest that they are not.

Comment 8: *“The predictions in Figure 1 should also include predictions for lower-frequency coherence, given what is discussed in Major point 1.”*

Reply: We agree with the reviewer that the effective connectivity results are if anything even more remarkable for theta impact and consistency across the patients than the other low frequencies. We revised the manuscript to ensure a more balanced presentation. Also, with the hypothesis Figure 1 we have now added the lower frequency bands.

Comment 9: *“Theta oscillations currently have a more ambiguous role in the predictive coding framework, having been associated with both feedforward entrainment to the speech syllabic rate, cognitive sampling at a theta rhythm and feedback processes “ -> I am not sure if theta oscillations would be considered to have an ambiguous role in inter-regional interactions. I suggest the authors consider some of the work cited in Major point 1 and consider these ideas in a little more detail.”*

Reply: We agree that theta oscillations have a clear and strong involvement in inter-regional interactions and have revised the manuscript to be clearer on this point. We have added the references from the list, which were relevant and useful as a link to the memory and cognition literature. We have also emphasized the strong impact on theta connectivity (lines 374-377).

Comment 10: *“The findings indicate that the left ATL is likely to act in two key ways: 1) as a higher level site in a constructive hierarchy” -> I am not sure how the results demonstrate a hierarchy here. IFG and other areas may play partially compensatory roles. How does this show a hierarchy?”*

Reply: We agree that IFG and other areas may work in a compensatory role, thus we have replaced “hierarchy” with “network” and “higher-level site” with “node” (lines 398-399).

Comment 11: *“Thereby, it is important for future studies to establish direct correspondence to source localized EEG or fMRI data in the same patients.” -> Scalp EEG, with a localization error using source reconstruction around 1cm³, cannot definitely identify deep brain signals. It is not clear how it could be used for those purposes here.”*

Reply: We recognize the spatial imprecision of scalp EEG signals, and the importance of site-specific intracranial recordings as a reference for EEG interpretation. We have revised this statement in the discussion recognizing this limitation: *“Although fMRI has reduced temporal precision and EEG has lower spatial specificity compared to intracranial EEG...”* (line 406).

References

- Carrera, E., & Tononi, G. (2014). *Diaschisis: past, present, future*. *Brain*, 137(Pt 9), 2408-2422. doi:10.1093/brain/awu101
- Finger, S., Koehler, P. J., & Jagella, C. (2004). *The Monakow concept of diaschisis: origins and perspectives*. *Arch Neurol*, 61(2), 283-288. doi:10.1001/archneur.61.2.283
- Henson, R. N., Greve, A., Cooper, E., Gregori, M., Simons, J. S., Geerligns, L., . . . Browne, G. (2016). *The effects of hippocampal lesions on MRI measures of structural and functional connectivity*. *Hippocampus*, 26(11), 1447-1463. doi:10.1002/hipo.22621
- Ray, S., & Maunsell, J. H. (2010). *Differences in gamma frequencies across visual cortex restrict their possible use in computation*. *Neuron*, 67(5), 885-896. doi:S0896-6273(10)00614-8 [pii] 10.1016/j.neuron.2010.08.004
- Shadlen, M. N., & Movshon, J. A. (1999). *Synchrony unbound: a critical evaluation of the temporal binding hypothesis*. *Neuron*, 24(1), 67-77, 111-125. doi:S0896-6273(00)80822-3 [pii]
- Solomon, E. A., Kragel, J. E., Sperling, M. R., Sharan, A., Worrell, G., Kucewicz, M., . . . Kahana, M. J. (2017). *Widespread theta synchrony and high-frequency desynchronization underlies enhanced cognition*. *Nat Commun*, 8(1), 1704. doi:10.1038/s41467-017-01763-2
- Vinck, M., van Wingerden, M., Womelsdorf, T., Fries, P., & Pennartz, C. M. (2010). *The pairwise phase consistency: a bias-free measure of rhythmic neuronal synchronization*. *Neuroimage*, 51(1), 112-122. doi:10.1016/j.neuroimage.2010.01.073
- Watrous, A. J., Tandon, N., Conner, C. R., Pieters, T., & Ekstrom, A. D. (2013). *Frequency-specific network connectivity increases underlie accurate spatiotemporal memory retrieval*. *Nature neuroscience*, 16(3), 349-356. doi:10.1038/nn.3315

Reply: Thank you for these very useful references, most of which have been added.

Additional references cited here:

- Baxendale, S., Thompson, P., Harkness, W. & Duncan, J. Predicting Memory Decline Following Epilepsy Surgery: A Multivariate Approach. *Epilepsia* 47, 1887–1894 (2006).
- Barnett, L. & Seth, A. K. Granger causality for state space models. *Phys. Rev. E* 91, 040101 (2015).
- Cheung, B. L. P., Riedner, B. A., Tononi, G. & Van Veen, B. D. Estimation of Cortical Connectivity From EEG Using State-Space Models. *IEEE Trans. Biomed. Eng.* 57, 2122–2134 (2010).
- Faes, L., Stramaglia, S. & Marinazzo, D. On the interpretability and computational reliability of frequency-domain Granger causality. *F1000Research* 6, 1710 (2017).
- Van de Steen, F. *et al.* Critical Comments on EEG Sensor Space Dynamical Connectivity Analysis. *Brain Topogr.* 32, 643–654 (2019).

Reviewer #3: *“The authors of the present manuscript tested the necessity of anterior temporal lobe (ATL) for semantic processing and its role as a “hub” in semantic processing. Two patients with intracranially implanted electrodes participated and EEG data were collected pre- and post- ATL-disconnection surgery with a specific focus on inferior frontal gyrus (IFG), Heschel's Gyrus (HG) and superior temporal sulcus (STS). Patients performed active and passive versions of a semantic prediction task in which participants listen to sentences that prime perception of a final target word. The authors find changes in spectral signals as well as connectivity in the comparison of pre- to post-surgical data. The authors state that their findings are consistent with the predictive coding framework, wherein ATL is responsible for providing predictions via feedback connections to downstream perceptual regions (e.g. HG) and that in the absence of ATL, other regions such as IFG engage with sensory regions differently to compensate. The question motivating the study -- necessity and role of ATL in semantic processing -- would be of interest to the field. However, the many methodological issues and conflicting results limit my enthusiasm for the manuscript and do not clearly support the conclusions. I detail these concerns below.”*

Reply. We sincerely appreciate the insightful comments and guidance on how to strengthen and clarify the manuscript. We have worked to address all of the suggested points, as follows.

Comment 1: *“Inconsistent findings that are challenging to interpret - Pre-surgery patient performance is not comparable to control performance, suggesting that semantic processing is already disrupted in these patients even before ATL disconnection. Thus, it is hard to conclude that the changes observed in the present study reflect how mechanisms might change following ATL loss in other contexts. Indeed, given the difference between patients and controls, patients may already be engaging compensatory mechanisms on account of their epilepsy, and thus the IFG changes observed are not a response to ATL loss specifically, but the combination of other resection (see below) and ATL disconnection. IFG changes may not be an “immediate” response if other deficits had arisen prior to ATL disconnection. The authors do note that this is a within-subjects case of cases study; however, the ability of the present findings to inform other work seems even more limited given this behavioral discrepancy.”*

Reply: This is a key point and a valid concern of studies in patients with epilepsy. In the revised manuscript we have strengthened this in two key ways, one directly in relation to the results including the new data and analyses of this study as part of the revision, and second in reference to prior studies on chronic re-organization of normal cognitive networks. We have also ensured that all neurophysiological results are directly of the acute pre- versus post-operative data, with the pre-operative data taking into account any deviation from normality in the pre-operative data. We hope this additional data and information help to minimize the concern and demonstrate that the approach and interpretations are based on consistent effects in both patients likely to generalize to others. We copy the following text from the revised manuscript here for convenience:

New section on *Minimizing Chronic Epilepsy Network Impact* (starting at line 858): *“The current study conducted direct pre- and post-disconnection comparison tens of minutes prior to and after the ATL disconnection procedure itself, with the pre-disconnection results acting as an important reference that incorporates any prior effects on the system. As with other studies in epilepsy patients the system cannot be considered normal. In relation to previous literature comparing the semantic networks of patients with chronic anterior temporal lobe epilepsy to healthy individuals using neuroimaging, Shimotake and colleagues (2015) show that the area of maximal semantic activation in patients and controls is in the same locus. However, it may be that the maximal location of activation is unchanged, but that there is already some compensation and more reliance on other nodes in the network, and on bilateral representations (see Schapiro et al., 2013 for an argument along these lines). This would likely reduce changes seen after ATL disconnection, as the epilepsy would have in essence already induced compensation or a partial disruption/disconnection of the functional network. Thus,*

here we may not be seeing all of the changes that might occur from disconnecting an ATL that had not experienced seizures. However, acute changes as reported here shortly before and after the ATL disconnection procedure likely reflect the acute loss of core processes that have not been disrupted by chronic epilepsy or its causes. Thereby, findings of immediate alterations in the neural network would have occurred even in the face of possible pre-operative chronic compensatory processes. Also, in these patients, the pre-surgical neurophysiological system shows fairly normal structural connectivity and function in both patients, including speech-related and mismatch responses in HG and speech-related responses in IFG (Fig. 3B). We also only make comparisons between pre resection and post resection effects that are significant and consistent in both patients, a within-subjects comparison, rather than comparisons to data in controls. Regarding the reported behavioral and neurophysiological results, the neurophysiological results are much more immediate pre- and post-surgical comparisons conducted tens of minutes before and after only the ATL disconnection step, whereas the behavioral results were only possible to obtain months before and after the neurosurgical procedure, which in one of the patients also involved the further MTL resection step. The pre-surgical neurophysiological responses serve as a more recent baseline comparison, taking into account any pre-existing network-level functional alteration and assessing only the additional impact of the loss of the ATL."

Revised Discussion Lines 378-393: "Although we were only able to study two patients here, there are several reasons to believe that these results are generalizable to other patients. Firstly, they are consistent across individuals despite significantly different epilepsy etiologies and timescales: P1 had hippocampal sclerosis and secondary gliosis of the temporal lobe, an acquired and progressive condition, while P2 had a congenital cavernoma present since birth. Secondly, our patients' neurophysiological system shows fairly normal structural connectivity pre-operatively, including speech-related and mismatch responses in HG and the expected pre-operative speech-related responses in IFG (Fig. 3B). Thirdly, previous literature comparing the semantic networks of patients with chronic anterior temporal lobe epilepsy to healthy individuals using neuroimaging (Figure 1 in Shimotake et al., 2015) shows that the area of maximal semantic activation in patients and controls is in the same locus, despite this overlapping with the epilepsy focus in the patients. Finally, we are making only comparisons between pre resection and post resection in the same individuals, rather than making any comparisons to connectivity in controls that might be confounded by pre-surgical connectivity changes as a result of epilepsy. Thereby, the overall findings suggest that the architecture of, impact on and forms of attempts at immediate compensation in these networks is well conserved despite pathology."

Comment 2: "The main effect of disconnection on the behavioral responses occurs in opposite directions across the two patients. The authors suggest that this may be due to additional medial temporal lobe resection in patient 1. At the very least, it seems that this cannot be described as a main effect of ATL disconnection. Why disconnection might improve performance is also challenging to understand. Finally, these findings, and the fact that one patient had a more extensive resection, suggest that pre-surgery performance and neural signals are unlikely to reflect typical neural processes or mechanisms."

Reply: We fully agree with the reviewer about the importance of care in reporting these two rare cases and in the revision have aimed to be careful throughout to not base interpretations on inconsistent findings. With the behavioral results, we agree that epilepsy patients are not likely to have a fully normal system that could affect memory and semantic functions for MTL epilepsy, which the revised manuscript now considers more carefully (see reply to Comment 1 above). We have also ensured that all comparisons are directly of the pre- versus post-operative data, with the pre-operative data being the baseline condition. This takes into account any deviation from normality in the pre-operative data, although as the reviewer notes there are important caveats with the behavioral data being possible to obtain months before and after the surgical procedure. Interestingly, the patients' results did indicate that both had a partial initial semantic context effect (floor /b/ bias

result, lines 146-149). However, the /p/ bias condition was well within the normative range for both patients. These are the ‘baseline’ conditions being compared to the post-operative data, and both patients were significantly impaired on the /p/ bias condition post-disconnection. The revised manuscript now also notes that unlike the *neurophysiological* recordings conducted pre- and post-operatively with only the ATL disconnection, we cannot rule out that the additional MTL resection step in patient P1 may have contributed to their behavioral semantic processing impairment post-operatively. However, this was not the case for P2 who did not have the additional MTL resection (lines 341-344), and the neurophysiological results are all obtained prior to the MTL resection in P1. We have integrated this additional information into the paper and checked that interpretations are based on consistency in both patients.

Furthermore, this reviewer and Reviewer 2 inspired us to analyze additional pre- and post-operative neuropsychological testing data. We have added this additional information to the Results (lines 180-191): “Post-operatively, both participants showed significant improvement in short-term memory for faces (WMS-III Faces I; RCI: P1: 3.402, P2: 2.91), which is not unexpected in MTL epilepsy patients (Baxendale et al., 2006). No other tests showed significant changes consistently across both participants at the time of testing. Two language-related tests were administered to both participants, however, only pre-operatively: the Boston Naming Test (BNT) which is a test for naming drawings of objects, and the Controlled Oral Word Association (COWA) Test, which is a verbal fluency test. On the latter, none of the participants were impaired pre-operatively, and, on the BNT, Participant 1 was right on the cusp of impairment (score of 48), while Participant 2 was in the average range (score of 55).

Comment 3: “The authors state, “one of the few neurophysiological signatures that remained largely unaffected pre- and post-disconnection in both participants was the preserved timing of the auditory cortical responses to the VOTs.” This claim is also repeated in the discussion. However, the authors report a significant effect for Peak 2 + patient 2. It is challenging to compare pre- vs. post-disconnection in Figure 3 as the ERPs are presented in separate panels.”

Reply: We appreciate the reviewer ensuring that also this interpretation is checked for accuracy and consistency across the two subjects. We agree that “largely unaffected” is not accurate given that Peak 2 is significantly different post-disconnection for P2. We have qualified the interpretation as “unaffected in both patients for the Peak 1 component” and “not consistent” for Peak 2 across the two participants (lines 245-251). We also provide the VOT data from the figures in Suppl. Table 3. Moreover, we have replotted the data so that the results can be visualized more clearly in Fig. 3A (for convenience shown next).

(A) Speech VOT responses largely intact post-disconnection

Comment 4: “Multiple methodological details that confound the results or lack clarity. - A number of experimental details are missing and as such it is challenging to understand how patients form expectations in the semantic prediction task. It appears that the same sentence is repeated 24 times in a session, as suggested by the text stating that 12 sentences were used over 288 trials. I would infer from “The target word was manipulated along a VOT continuum with 6 steps” that there were

four trials each with each stepped VOT, but this was not made explicit. If that is the case, why would participants predict the semantically congruent word? Especially considering that there were multiple sessions with what appears to be the same stimuli (since the pre-surgery sessions were used to select stimuli for the surgery sessions), participants should learn to expect other words than the semantically congruent one. In the absence of predictions, and possibly when predictions are modified by experimental parameters, the role of a putative hub region such as ATL will necessarily be different than it would be for more naturalistic semantic predictions.”

Reply: We now provide the detailed information on the semantic prediction task. We have added the additional information about the study design in Suppl. Table 5 showing all of the sentences and conditions. In the study design, filler trials were included to ensure that the participants' expectations were fulfilled on the majority of the trials, and that participants would expect the semantically congruent word. Furthermore, the catch trials (in the pre- and post-operative testing) were used to ensure that the participants engaged with the semantic information in the sentences and generated meaning-based expectations throughout the task for the target word (also see Sarrett et al., 2020). We have also cited published prediction trials demonstrating that lifelong learned associations and semantic priming cannot be easily overridden by immediate experimental context (e.g., Sohoglu et al., 2012; Blank et al. 2018), and that mismatch responses may be attenuated in a context where incongruence is frequently occurring but are difficult to abolish (Näätänen, 2004). We have made these points clearer in the revised manuscript (lines 566-569 and lines 211-219).

Comment 5: *“Putting aside the above issue, if patients are predicting the semantically congruent word prior to target presentation, using the time prior to target presentation as a baseline is inappropriate. Patients should be making predictions during this time interval, it is not “neutral” (i.e. it is not a true baseline interval, if one exists) and will artificially create effects during the target interval. This impacts both the selection of the “speech responsive” electrodes and pre/post surgery comparisons. An appropriate pre-stimulus baseline would be a time interval before a trial (sentence) begins, when patients have no expectations/preparation. Alternatively, the data could be z-scored across all trials, which would obviate the need for baseline interval selection (which appears to be what the authors did for the state-space conditional granger causality analysis). “*

Reply: We also were concerned that the pre-target word baseline choice and de-meaning the baseline period before it might minimize semantic prediction effects. However, there is little evidence for a prediction signal during the baseline, which can be seen as a not statistically significant mismatch response during the baseline period in Fig. 3C before time 0 when the target word starts. There are some natural fluctuations during the baseline period for both the pre- and post-operative target word response data, but none are significantly different pre- and post-operatively during the baseline. As suggested, we evaluated having the baseline before the start of the sentence, but that baseline was too far removed from the speech word and the neurophysiological signal power fluctuations too great for this to be an appropriate baseline for the later target word response period. Moreover, our target word mismatch responses are very similar to prior reports regarding both the mid incongruency (greater than congruency) and later congruency (greater than incongruency) components (e.g., compare our Fig. 3C to Fig. 7B in Cope et al., 2017). Also, a recent decoding study of prediction-related effects in rodent auditory cortex was unable to decode the prior sounds during a silent gap, but decoding was successful when a broadband noise was presented during this period (Cappotto et al., 2022). Thus, many studies, including ours, tend to find prediction-related effects during the sensory stimulation period, or at least these can be more substantial than those during silent periods. We have also clarified the basis for the analyses that z-scoring is required for analyses like the CGC where mean centering and stationarity are followed by statistical testing (permutation and FDR corrected tests). Also, in showing the pre- and post-disconnection evoked potentials, we find it more informative and important for the reasons noted by the reviewer to stay close to the raw data by displaying the de-

measured evoked potential alongside the results from the cluster-corrected statistical tests of the neurophysiological response pre- and post-disconnection.

Comment 6: *“There is the potential that changes in pre- vs. post-surgery signals are driven by the selection of different electrodes in each condition. “*

Reply: We apologize for the lack of clarity in the prior Methods on the procedure for selecting speech responsive contacts, which is based on *only the pre-operative data*. We have revised the Methods to ensure the required detail is available that the HG and STG electrodes stayed in the same location during the ATL disconnection procedure. The IFG recording grid, however, had to be removed by the neurosurgeon before the ATL disconnection procedure and replaced afterwards. We used protocolized procedures to re-register the pre- and post-disconnection recording contacts, similar to an MRI nearest neighbor co-registration approach. Our protocol and registration steps are detailed in the Methods (lines 710-741).

Comment 7: *“No justification is provided for frequency band selection and bands do not match across analyses (Figure 3 vs. Figure 5). The only written definition of the bands appears at the end of the methods. In general, the authors make broad statements about frequency that are not clearly motivated (e.g. “beta/alpha < 30 Hz” on page 5; multiple frequencies fall below 30 Hz). It is unclear why the authors expected strong broadband activity to speech sounds (Page 8), especially considering evidence for high frequency increases and low frequency decreases during auditory perception (Crone et al. (2001) Induced electrocorticographic gamma activity during auditory perception. Clinical Neurophysiology). “*

Reply: As suggested, we have replaced “broadband activity to speech sounds” with a much more nuanced treatment of the speech responses (lines 252-263) to ensure the text more accurately reflects both the alpha suppression and the increased high gamma response (Fig. 3B), similar to reported speech responses with iEEG recordings (e.g., Crone et al. 2001). The frequency bands only mismatch in the case of the CGC results, because no significant CGC effects were identified above 50 Hz. Therefore, we did not include >50Hz frequency band results in the CGC results figures. This methods figure was also not crucial to interpretation and was moved to Suppl. Fig. 7.

Comment 8: *“More information is needed as to how centrality, hubness, and Conditional Granger Causality strength are determined. There should be explicit values provided, based on prior work, that must be observed for a region to be defined as “hub-like.” How regions of interest (ROIs) were defined (i.e. how electrode inclusion in an ROI was determined) and the number of electrodes per ROI is not reported. It is unclear why both patients’ data were concatenated seeing as patients performed differently on the behavioral measures. “*

Reply: We have expanded the information on how centrality, hubness and CGC strength are determined both in the Results (lines 202-210) and in a new section in the Methods (starting at line 826). Under the definition, hubness is a continuous variable. However, as the reviewer highlights, instead of defining an arbitrary threshold for whether we consider a given node to be a hub, we statistically tested the claim about the hubness of the TP in the data shown in Figure 3D, as follows: Significance testing was calculated by randomly swapping edges of the directed weighted networks generating 10,000 surrogate networks to serve as an empirical null distribution for the sum of in- and outflow, which can then be used to compute p-values for likelihood of the hubness data from the five nodes representing chance. As noted in the manuscript (lines 208-210) only the TP contacts reached significance with this test ($p = 0.018$) in the gamma band; for further results, see Suppl. Fig. 6. We also report how the ROIs were defined and the number of electrodes in each ROI (lines 710-741) and have ensured that main interpretations throughout are based on consistent effects in both patients.

Comment 9: “Logic and motivation of the hypothesis; interpretation of the findings. The authors propose that IFG may fill a compensatory role, “Other regions could compensate for an impact on the ATL, or can maintain established semantic representations, at least temporarily. Candidate compensatory regions would include the language-related hub in the left inferior frontal gyrus (IFG)” (Page 2). However, if IFG is performing a compensatory role, why would feedback signals (alpha/beta) be disrupted after ATL disconnection (Figure 1)? Shouldn’t feedback signals increase if IFG is compensating for the loss of ATL? Broadly speaking, the hypotheses feel a bit post-hoc and although the prediction framework makes sense on its own, the specific predictions made (i.e., disruptions to feedback signals) do not seem to match this hypothesis. Likewise, the findings do not clearly support the predictive coding framework, given the issues with prediction outlined above.”

Reply: We agree with the reviewer on these astute observations, particularly that our initial hypothesis of likely brain areas to compensate for the impact of the loss of the ATL would include the IFG. However, consistent with the important role of the ATL as a hub that cannot be completely compensated for, even by the IFG, the remaining recorded nodes in this network including the IFG showed substantial disruption. The revised manuscript Discussion (starting at lines 321) now notes that the data do not support the complete disruption hypothesis, but instead support the modern diaschisis hypothesis (motivated by Reviewer 2) generally, and more specifically, the predictive coding hypothesis in three key forms (see revised Discussion lines 324-334). First, HG gamma magnification to speech sounds is expected by the framework as the result of losing the prediction signal. Indeed, some readers might view this as the resulting impact rather than compensation after the loss of the ATL and its top-down signals on earlier stages of auditory cortex. However, by the framework in which a top-down prediction is inaccurate or disrupted, the prediction error is a learning signal that aims to update internal models in hierarchically higher nodes. Second, also consistent with the predictive coding framework, is the observed disruption of the mismatch response to the target word. Notably, the pre-disconnection mismatch response components to the target word are very similar to those reported elsewhere (compare Fig. 3C with Fig. 7B in Cope et al., 2017). The third facet is the magnification of effective connectivity. Although we also expected this to involve the IFG, that was only the case for one of the patients, whereas for the other such magnification in effective connectivity instead involved the STG, which we interpret as incomplete attempts of the neural network to compensate, again underscoring the importance of the loss of the ATL. We have also noticed that the prior title could be misinterpreted as ‘complete’ compensation, which we never intended. Thereby we have ensured that even the title is accurate and consistent with the results: *Immediate Neural Impact and Incomplete Compensation After Semantic Hub Disconnection*.

We sincerely appreciate all of the reviewers’ time and insightful comments, and we hope that the reviewers find that following the peer-review process the revised manuscript has now achieved the required accuracy in interpretation of results consistent across both patients.

Additional references cited here:

Baxendale, S., Thompson, P., Harkness, W. & Duncan, J. Predicting Memory Decline Following Epilepsy Surgery: A Multivariate Approach. *Epilepsia* **47**, 1887–1894 (2006).

Blank, H. & Davis, M. H. Prediction Errors but Not Sharpened Signals Simulate Multivoxel fMRI Patterns during Speech Perception. *PLOS Biol.* **14**, e1002577 (2016).

Cope, T. E. *et al.* Evidence for causal top-down frontal contributions to predictive processes in speech perception. *Nat. Commun.* **8**, 2154 (2017).

Cappotto, D., Kang, H., Li, K., Melloni, L., Schnupp, J., Auksztulewicz, R. Simultaneous mnemonic and predictive representations in the auditory cortex. *Curr. Biol.* **32**, 2548-2555.e5 (2022).

Crone, N.E., Boatman, D., Gordon, B., Hao, L. Induced electrocorticographic gamma activity during auditory perception. *Clin. Neurophys.* **112**, 565-582, (2001).

Näätänen, R., Pakarinen, S., Rinne, T. & Takegata, R. The mismatch negativity (MMN): towards the optimal paradigm. *Clin. Neurophysiol. Off. J. Int. Fed. Clin. Neurophysiol.* **115**, 140–144 (2004).

Sarrett, M. E., McMurray, B. & Kapnoula, E. C. Dynamic EEG analysis during language comprehension reveals interactive cascades between perceptual processing and sentential expectations. *Brain Lang.* **211**, 104875 (2020).

Schapiro, A. C., McClelland, J. L., Welbourne, S. R., Rogers, T. T. & Lambon Ralph, M. A. Why Bilateral Damage Is Worse than Unilateral Damage to the Brain. *J. Cogn. Neurosci.* **25**, 2107–2123 (2013).

Shimotake, A. *et al.* Direct Exploration of the Role of the Ventral Anterior Temporal Lobe in Semantic Memory: Cortical Stimulation and Local Field Potential Evidence From Subdural Grid Electrodes. *Cereb. Cortex* **25**, 3802–3817 (2015).

Sohoglu, E., Peelle, J. E., Carlyon, R. P. & Davis, M. H. Predictive Top-Down Integration of Prior Knowledge during Speech Perception. *J. Neurosci.* **32**, 8443–8453 (2012).

REVIEWER COMMENTS

Reviewer #1 (Remarks to the Author):

I felt the authors did a good job responding to my points. But I do have one lingering critical point.

Regarding the author's response to my first main point, was there a correction for multiple comparisons across region and frequency with the analysis for Figure 3D where they found an effect for only the gamma band at $p=0.018$? If this effect was not corrected for multiple comparisons, then it would seem to be a substantial problem because this was a rather exploratory sort of analysis. I think it would be important overall that the paper provides a strong statistical justification for the claims related to Figure 3D.

Reviewer #2 (Remarks to the Author):

The manuscript has been substantially revised and now sits on much firmer ground statistically and scholarly. I am still finding the behavioral results a little hard to follow for P1 vs. P2: it looks like P1 and P2 show quite different biases in Figure 2E and it would be helpful to have those radical differences in biases explained in a little more detail. Also, Figure 3A (right panel) is hard to follow. Some of this may be the color scheme, which is almost identical for pre peak 1 and pre peak 2 and post peak 1 and post peak 2. It would help to have the lines better differentiated and the meaning of the figure explained in a little more detail.

Reviewer #3 (Remarks to the Author):

The authors have addressed a majority of the concerns that I raised previously and the manuscript is overall very much improved. I have two remaining concerns, one major and one minor.

Major. I previously raised concerns about the use of the pre-target period as a baseline. The authors state that the absence of a pre-target mismatch dissociation means that the pre-target period is a suitable baseline period; however, this does not address the issue. Participants should generate a prediction on all trials, both match/congruent and mismatch/incongruent. This means that there should be a prediction signal on all trials that would be subtracted off of the target locked data. It appears that Cope et al., 2017 used the same type of baseline period. Finding that the currently reported results are the same if a z-scoring approach is used in lieu of a pre-target baseline period will address this concern.

Minor. The right panel of Figure 3A has made the ERP comparisons clearer, though I would recommend changing the line colors as the instinct is to compare pre Peak 1 to pre Peak 2 since both are shades of blue. Having both Peak 1 lines within the same color family and the Peak 2 in their own different color family will enable readers to easily make the intended comparison between pre and post.

Reviewer Point-by-point Replies

Reviewer #1

Comment 1: *"I felt the authors did a good job responding to my points. But I do have one lingering critical point. Regarding the author's response to my first main point, was there a correction for multiple comparisons across region and frequency with the analysis for Figure 3D where they found an effect for only the gamma band at $p=0.018$? If this effect was not corrected for multiple comparisons, then it would seem to be a substantial problem because this was a rather exploratory sort of analysis. I think it would be important overall that the paper provides a strong statistical justification for the claims related to Figure 3D."*

Reply: We appreciate the opportunity to explicitly formalize and test the *a priori* hypothesis that the temporal pole (TP) acts as a functional hub with multiple comparisons correction (lines 212-214). The hypothesis is firmly grounded in the prior literature, which, although not necessary to re-test here, we evaluated with the results from the present study. When the hypothesis is statistically tested (Methods, lines 841-856), we find that the TP in the gamma band is significantly stronger as a functional hub than the other regions even when corrected for multiple comparisons ($p(\text{corrected}) = 0.036$, Benjamini-Hochberg procedure to control the false discovery rate at a level of 0.05). The results are described in lines 214-216, and referenced in the legends to Fig. 3D, Suppl. Fig. 7 and shown in Suppl. Table 6.

Reviewer #2

Comment 1: *"The manuscript has been substantially revised and now sits on much firmer ground statistically and scholarly. I am still finding the behavioral results a little hard to follow for P1 vs. P2: it looks like P1 and P2 show quite different biases in Figure 2E and it would be helpful to have those radical differences in biases explained in a little more detail."*

Reply: We agree that these behavioral results were not previously explained clearly, and have rewritten the description of Figure 2E in the Results section (lines 146-160):

"Figure 2E shows that, before surgery, both participants' phoneme categorization functions displayed an initial semantic context effect that closely mirrored the average response to the /p/ biased condition in the control participants, with both blue dashed lines largely overlapping with the thick black line (controls). Two months after surgery, both participants had a much-reduced effect of voice onset time, manifest in flatter red dashed lines. Participant 1 adopted a strategy across both conditions (c.f. Suppl. Fig. 1) that was more heavily based upon preceding sentential context, with an overall shift upwards towards the /p/ bias. Conversely, Participant 2 retained some ability to integrate voice onset time, but this was diminished from the pre-surgical state, and they showed a new bias in the opposite direction towards /b/, incongruent with the preceding sentential context. This bias cannot be easily explained as a response button bias (the patient just pushing one of the response buttons) given their good performance on the control tasks. Overall, while the post-surgical biases of the two patients on the semantic priming task appear quite different, they both represent a loss of adaptive function. This is analogous to the failure of comprehension resulting from overly precise perceptual predictions in patients with damage to frontal lobe language regions (Cope et al., 2017; 2023)."

The overall theme is that resection of the temporal pole reduced the ability of both patients to adaptively integrate semantic expectations from sentential context with voice onset time. We do not make strong claims about the precise nature of the behavioral consequence of unilateral temporal pole resection, particularly months later when there is likely to be at least partial compensation from the contralateral side. All we wish to illustrate is that there is a lasting change in behavior, in an auditory semantic task, underlining the importance of the neurophysiological changes that provide the primary results in the manuscript.

Comment 2: “Also, Figure 3A (right panel) is hard to follow. Some of this may be the color scheme, which is almost identical for pre peak 1 and pre peak 2 and post peak 1 and post peak 2. It would help to have the lines better differentiated and the meaning of the figure explained in a little more detail.”

Reply: We have now made the Peak 1 and Peak 2 results of the same color family (Peak 1 – blue; Peak 2 – green), but clearly distinct. We have also added the interpretation of the results to the Fig. 3A legend (summarizing the Results text, line 257).

References:

Cope, T. E., Sohoglu, E., Sedley, W., Patterson, K., Jones, P. S., Wiggins, J., ... & Rowe, J. B. (2017). Evidence for causal top-down frontal contributions to predictive processes in speech perception. *Nature Communications*, 8(1), 2154.

Cope, T. E., Sohoglu, E., Peterson, K. A., Jones, P. S., Rua, C., Passamonti, L., ... & Rowe, J. B. (2023). Temporal lobe perceptual predictions for speech are instantiated in motor cortex and reconciled by inferior frontal cortex. *Cell Reports*, 42(5).

Reviewer #3

Comment 1: “*The authors have addressed a majority of the concerns that I raised previously and the manuscript is overall very much improved. I have two remaining concerns, one major and one minor. Major. I previously raised concerns about the use of the pre-target period as a baseline. The authors state that the absence of a pre-target mismatch dissociation means that the pre-target period is a suitable baseline period; however, this does not address the issue. Participants should generate a prediction on all trials, both match/congruent and mismatch/incongruent. This means that there should be a prediction signal on all trials that would be subtracted off of the target locked data. It appears that Cope et al., 2017 used the same type of baseline period. Finding that the currently reported results are the same if a z-scoring approach is used in lieu of a pre-target baseline period will address this concern.*”

Reply: We have worked to address this point with additional analyses. The results of these analyses show that the mismatch response results are robust and not dependent on the baseline period selection. In the new Suppl. Fig. 6, we work through a series of control questions and analyses as explained next:

Does having no baseline correction at all alter the results? Panel A replicates the figure from the mismatch response results as a point of reference for the additional analyses. This original analysis used the 150 ms pre-target word period only for de-meaning the signals around the baseline. Panel B shows the results without any baseline correction, and this makes no substantial impact on the results and their significance. Also, the sentences used in our experiment were—by design—balanced before the target word to elicit consistent prediction effects before the target word was heard. We agree that the participants are generating a prediction during this interval, however, the difference between congruent and

incongruent words can only be appreciated when the target word is heard, not before. We did not find any evidence in the results (Fig. 3C; Suppl. Fig. 6) to support the notion that there are differential prediction signals in the pre-target period. Specifically, not applying any baseline correction to the data does not introduce any significant differences on the baseline period prior to the congruent and incongruent words.

Do the results differ if the pre-sentence period is used as a baseline? Panel C shows that using the 150 ms period prior to the start of the sentence does not substantially alter the results, which continue to show the strong pre-disconnection mismatch response being substantially reduced post-disconnection. We would also like to point out that our paradigm uses sentences of variable length, and in most cases the selected pre-sentence baseline period is far removed from the time window of interest, yet the key observation remains.

Do the results differ if z-scoring is used? In Panel D, below we created a z-scored version of the data analysis, taking the mean of the entire epoch, subtracting it from each time point in a given trial, and then dividing by the standard deviation of the epoch (trial-by-trial). This is the only method of z-scoring that we believe obviates the need for baseline selection, however this approach does not make sense for ERP analysis where there are expected differences in signal magnitude between conditions. If the signal to noise ratio is relatively high, then any epoch with an evoked response will have a larger standard deviation than an epoch without an evoked response, meaning that it is downweighted in the averaging. In our case, where evoked responses are significantly attenuated by the anterior temporal lobe disconnection, the result of z-scoring based on the standard deviation of the epoch is an unnecessary downscaling and increased variation of the pre-disconnection data. Consequently, as can be seen in Suppl. Fig. 6D, the form of the post-disconnection waveform is largely identical between panels B and D, while the pre-disconnection waveform is flattened out. We would also like to add that z-scoring with respect to individual trials will always lead to a different scaling of each trial. This contradicts the basic tenet of ERP analysis, which is that summing together deflections of equivalent magnitude, all of which are masked by noise, is supposed to lead to the summation of the ERP wave, while the overlying noise gets averaged out. If each trial is scaled individually, this no longer is the case, therefore z-scoring on a trial-by-trial basis is inappropriate for these data. On the other hand, if we were to apply z-scoring to the overall ERP, this would only scale the y-axis to z-values without altering waveform morphology or statistics. Therefore, z-scoring on this dataset is inappropriate and does not seem to be in line with the reviewer's intention that it should increase the signal-to-noise ratio. So, while we show Suppl. Fig. 6D here, in the manuscript version we propose only to include panels A-C.

We use an analysis approach that is common for ERP analyses: It is worth pointing out that our approach is commonly used in the ERP community (Sarrett et al., 2020; Getz and Toscano, 2019; Cope et al., 2017; Cohen, 2014; Näätänen et al., 1993; Bodatsch et al., 2011; Lee et al. 2017; Kimura et al., 2009). Some form of baseline correction is almost always applied to EEG data when creating event-related potentials to reduce the effects of temporal drifts, and to remove low frequency components (Cohen, 2014). Luck's seminal textbook on the fundamentals of ERP analysis is of particular relevance here (Luck, 2014, chapter 8). Luck states that "it is almost always necessary to use the average voltage during the prestimulus interval as a baseline". Furthermore, in the same chapter after describing a paradigm much like ours, Luck states "*Although this baseline is contaminated by overlap from the previous word, this overlap is not usually a problem if it is identical across conditions... Consequently, it may be worth using a baseline that is contaminated by overlap if it allows the baseline period*

to be closer in time to the measurement period.” The effectiveness and usefulness of classical baseline correction has certainly been and continues to be scrutinized (e.g., Alday, 2019; Delorme, 2023), and Luck, too, points out its caveats (2014, Chapter 8), but its general use does not appear to be diminishing (c.f., Kovacs et al., 2023; Sarasso et al., 2022; Nourski et al., 2021; Jacobsen et al., 2021, etc.).

Taking all of these results together, we can confidently say that the no baseline correction and the pre-sentence baseline correction methods recapitulate the effects we have seen in the original analysis using the pre-target word period for baseline correction. We believe the additional analyses help to justify our choice of analysis and demonstrate the robustness of the results to no- or an alternative-baseline correction. The manuscript now notes these additional analyses (lines 277-278) and provides the additional baselining results in Suppl. Fig. 6.

A) Original, pre-target word baseline corrected

B) No baseline correction applied

C) Pre-sentence period baseline corrected

D) Z-score, trial by trial

— Average stimulus length — Pre-disconnection — Post-disconnection
— P 1 Significant ($p < 0.05$) — P 2 Significant ($p < 0.05$)

Mismatch response evaluated with different baselining methods. Panel A replicates Fig. 3C to compare to results with other baselining approaches (for this result the canonical baseline correction was used during the pre-target word period (-150 to 0 ms, as per target word onset) in each trial separately for the congruent and incongruent words. B) No baseline correction is applied to the epochs, which shows similar results to the

canonical baseline correction. C) Pre-sentence baseline correction: using the baselining method described in A, but on the interval taken from before the onset of the biasing sentence (-150 to 0 ms with respect to sentence onset). The // marks a discontinuity in the x axis, showing that the left of this represents the times taken from before the sentence onset, and the times after this represent the times after target word onset. D) Trial-by-trial z-scoring: taking the mean of the entire epoch for each trial, subtracting the signal for each trial, and then dividing by its standard deviation. This analysis, however, although not reliant on a baseline, over-inflates the trial-by-trial noise and greatly reduces any signal obtained by downweighting epochs with large evoked responses. Figure format: blue traces show the pre-disconnection difference wave between the congruent and incongruent event-related potentials, red traces show the post-disconnection difference wave. The black line denotes the average target word length, the grey lines show the significant period based on the cluster-based permutation testing ($p < 0.05$, 10 000 permutations) for the two participants, separately. The cyan and magenta asterisk point to effects that are either magnified or squashed by the trial-by-trial z-scoring method when compared to the original baselining method (compare panel D to A).

Comment 2: “Minor. The right panel of Figure 3A has made the ERP comparisons clearer, though I would recommend changing the line colors as the instinct is to compare pre Peak 1 to pre Peak 2 since both are shades of blue. Having both Peak 1 lines within the same color family and the Peak 2 in their own different color family will enable readers to easily make the intended comparison between pre and post.”

Reply: We have made the Peak 1 and Peak 2 results of the same color family (Peak 1 – blue; Peak 2 – green). We have also added the interpretation of the results to the Fig. 3A legend (summarizing the Results text, line 254).

References:

- Alday, P. M. (2019). How much baseline correction do we need in ERP research? Extended GLM model can replace baseline correction while lifting its limits. *Psychophysiology*, 56(12), e13451.
- Bodatsch, M., Ruhrmann, S., Wagner, M., Müller, R., Schultze-Lutter, F., Frommann, I., ... & Brockhaus-Dumke, A. (2011). Prediction of psychosis by mismatch negativity. *Biological psychiatry*, 69(10), 959-966.
- Cohen, M. X. (2014). *Analyzing neural time series data: theory and practice*. MIT press.
- Cope, T. E., Sohoglu, E., Sedley, W., Patterson, K., Jones, P. S., Wiggins, J., ... & Rowe, J. B. (2017). Evidence for causal top-down frontal contributions to predictive processes in speech perception. *Nature Communications*, 8(1), 2154.
- Delorme, A. (2023). EEG is better left alone. *Scientific reports*, 13(1), 2372.
- Getz, L. M., & Toscano, J. C. (2019). Electrophysiological evidence for top-down lexical influences on early speech perception. *Psychological science*, 30(6), 830-841.
- Jacobsen, T., Bäß, P., Roje, A., Winkler, I., Schröger, E., & Horváth, J. (2021). Word class and word frequency in the MMN looking glass. *Brain and Language*, 218, 104964.
- Kimura, M., Schröger, E., Czigler, I., & Ohira, H. (2010). Human visual system automatically encodes sequential regularities of discrete events. *Journal of Cognitive Neuroscience*, 22(6), 1124-1139.
- Kovács, P., Tóth, B., Honbolygó, F., Szalárdy, O., Kohári, A., Mády, K., ... & Winkler, I. (2023). Speech prosody supports speaker selection and auditory stream segregation in a multi-talker situation. *Brain Research*, 148246.
- Lee, M., Sehatpour, P., Hoptman, M. J., Lakatos, P., Dias, E. C., Kantrowitz, J. T., ... & Javitt, D. C. (2017). Neural mechanisms of mismatch negativity dysfunction in schizophrenia. *Molecular psychiatry*, 22(11), 1585-1593.
- Luck, S. J. (2014). *An introduction to the event-related potential technique*. MIT press.
- Näätänen, R., Paavilainen, P., Titinen, H., Jiang, D., & Alho, K. (1993). Attention and mismatch negativity. *Psychophysiology*, 30(5), 436-450.
- Nourski, K. V., Steinschneider, M., Rhone, A. E., Mueller, R. N., Kawasaki, H., & Banks, M. I. (2021). Arousal state-dependence of interactions between short-and long-term auditory novelty responses in human subjects. *Frontiers in human neuroscience*, 15, 737230.
- Sarasso, P., Neppi-Modona, M., Rosaia, N., Perna, P., Barbieri, P., Del Fante, E., ... & Ronga, I. (2022). Nice and easy: Mismatch negativity responses reveal a significant correlation between aesthetic appreciation and perceptual learning. *Journal of Experimental Psychology: General*, 151(6), 1433.
- Sarrett, M. E., McMurray, B., & Kapnola, E. C. (2020). Dynamic EEG analysis during language comprehension reveals interactive cascades between perceptual processing and sentential expectations. *Brain and language*, 211, 104875.

REVIEWERS' COMMENTS

Reviewer #1 (Remarks to the Author):

The authors have adequately addressed my concern. I am glad that the effect remained significant after multiple comparison correction.

Reviewer #3 (Remarks to the Author):

The authors have addressed all of my concerns.

RESPONSE TO THE REVIEWERS' COMMENTS

Reviewer #1 (Remarks to the Author):

The authors have adequately addressed my concern. I am glad that the effect remained significant after multiple comparison correction.

Answer: We thank the reviewer for their time and effort in reviewing the manuscript.

Reviewer #3 (Remarks to the Author):

The authors have addressed all of my concerns.

Answer: We thank the reviewer for their time and effort in reviewing the manuscript.